# Unexpected doping effects on phonon transport in quasi-one-dimensional van der Waals crystal TiS$_3$ nanoribbons

Chenhan Liu ⓘ[1,2,6], Chao Wu[1,6], Xian Yi Tan[3,4,6], Yi Tao[1], Yin Zhang[1], Deyu Li ⓘ[5], Juekuan Yang ⓘ[1] ✉, Qingyu Yan ⓘ[3] ✉ & Yunfei Chen ⓘ[1] ✉

Doping usually reduces lattice thermal conductivity because of enhanced phonon-impurity scattering. Here, we report unexpected doping effects on the lattice thermal conductivity of quasi-one-dimensional (quasi-1D) van der Waals (vdW) TiS$_3$ nanoribbons. As the nanoribbon thickness reduces from ~80 to ~19 nm, the concentration of oxygen atoms has a monotonic increase along with a 7.4-fold enhancement in the thermal conductivity at room temperature. Through material characterizations and atomistic modellings, we find oxygen atoms diffuse more readily into thinner nanoribbons and more sulfur atoms are substituted. The doped oxygen atoms induce significant lattice contraction and coupling strength enhancement along the molecular chain direction while have little effect on vdW interactions, different from that doping atoms induce potential and structural distortions along all three-dimensional directions in 3D materials. With the enhancement of coupling strength, Young's modulus is enhanced while phonon-impurity scattering strength is suppressed, significantly improving the phonon thermal transport.

Benefited from actively manipulating electron transport in various semiconductor materials, microelectronics has been a driver for innovation in a wide range of applications since the invention of transistors in 1959. It is expected that atomic thin transistors made of low-dimensional materials help extend Moore's law. In the recent two decades, tremendous efforts have been made to explore two-dimensional (2D) van der Waals (vdW) materials and their potential applications in electronic[1,2], optoelectronic[3–5], bioelectronic[6], thermal[7,8], and energy storage devices[9–11]. These studies suggest that 2D materials provide atomic flat surfaces free from defects, in which electrons are less prone to be scattered and charges can flow relatively freely. Compared to 2D vdW materials, quasi-1D vdW materials[12] are not well explored, which can help extend Moore's law as well.

Quasi-1D vdW materials are composed of molecular/atomic chains with strong intrachain covalent or ionic bonds assembled through relatively weak interchain vdW interactions. Compared to the restriction of strong covalent bonds on the structure in 3D materials, the presence of the vdW gaps in quasi-1D vdW materials facilitates more freedom of structure manipulation and modification such as disassembly/reassembly[12,13], ions intercalation[14], and substitution[15]. For example, quasi-1D Ta$_2$Pt$_3$Se$_8$ and Ta$_2$Pd$_3$Se$_8$ nanowires can be assembled together to form nanoscale heterojunctions[13]. The physical

¹Jiangsu Key Laboratory for Design and Manufacture of Micro-Nano Biomedical Instruments, School of Mechanical Engineering, Southeast University, Nanjing 211100, P. R. China. ²Micro- and Nano-scale Thermal Measurement and Thermal Management Laboratory, School of Energy and Mechanical Engineering, Nanjing Normal University, Nanjing 210046, P. R. China. ³School of Materials Science and Engineering, Nanyang Technological University, 50 Nanyang Avenue, 639798 Singapore, Republic of Singapore. ⁴Institute of Materials Research and Engineering (IMRE), Agency for Science, Technology and Research (A*STAR), 2 Fusionopolis Way, Innovis #08-03, 138634 Singapore, Republic of Singapore. ⁵Department of Mechanical Engineering, Vanderbilt University, Nashville, TN 37235-1592, USA. ⁶These authors contributed equally: Chenhan Liu, Chao Wu, Xian Yi Tan. ✉e-mail: yangjk@seu.edu.cn; alexyan@ntu.edu.sg; yunfeichen@seu.edu.cn

properties of quasi-1D vdW materials are expected to be different from those of 3D materials because of the weak vdW interactions and the vdW gaps.

An interesting quasi-1D vdW crystal is titanium trisulfide (TiS$_3$)[16,17] and it is an *n*-type semiconductor with a bandgap of ~1 eV[18,19]. TiS$_3$ nanoribbons manifest higher current-carrying capacity than that of copper via field-effect gating[20]. Besides tunable electrical properties, excellent photoresponse up to 2910 A/W has also been observed in TiS$_3$ nanoribbon-based field-effect transistors[21]. The wide-range tunable electron mobility and ultrahigh optical responsivity[22] promise TiS$_3$ nanoribbons various applications in electronic and optoelectronic devices[23]. For device applications, thermal properties are critical for thermal management because device lifetime decays exponentially with temperature[24]. However, so far, the thermal properties of TiS$_3$ nanoribbons are not well explored[25-27].

In this work, doping effects on phonon transport in quasi-1D vdW crystal TiS3 nanoribbons are reported, which demonstrates that O atoms can easily diffuse into thin TiS$_3$ nanoribbons and significantly enhance rather than reduce lattice thermal conductivity. The replacement of sulfur atoms with oxygen atoms results in significant lattice contraction and coupling strength enhancement of TiS$_3$ along the molecular chain direction, with little effect on vdW strengths. Compared with the doped atom inducing strain and reducing lattice thermal conductivity in 3D materials, the significant lattice contraction enhances Young's modulus along the molecular chain direction in thin TiS$_3$ nanoribbons, which results in an enhanced phonon group velocity and suppressed phonon-impurity scattering strength. The combination effects lead to a 7.4-fold enhancement in thermal conductivity at room temperature. This work provides a new method to actively control phonon thermal transport through doping low-dimensional materials with small atoms.

## Results

### Thermal conductivity of TiS$_3$ nanoribbon

The TiS$_3$ crystal structure is schematically shown in Fig. 1a, b: (i) transition metal Ti atoms form trigonal prismatic coordination with S atoms through Ti-S bond; (ii) these trigonal prisms connect into quasi-

1D molecular chains through S-Ti-S bond along the *b*-axis direction; (iii) stacking of these quasi-1D chains through vdW interactions in the other 2D directions (*a*- and *c*-axis) into a quasi-1D vdW structure. Figure 1c shows a TiS$_3$ nanoribbon placed on a microdevice composed of two suspended membranes with integrated heaters and resistance thermometers serving as the heat source and heat sink, respectively. The thermal conductivity is extracted through a microthermal bridge scheme and additional platinum electrodes patterned on the suspended membranes that allow for four-probe measurements of the electrical conductivity. The dimensions of each measured nanoribbon sample are listed in Table 1. High-resolution transmission electron microscopy (HRTEM) images viewed along the [001] (Fig. 1d) and [100] directions (Fig. 1e) indicate that the nanoribbon is grown along the *b*-axis direction. The inset in Fig. 1d shows a fast Fourier transform pattern of the HRTEM image, from which the lattice constants of all samples are extracted and listed in Table 2.

Figure 2a plots the measured thermal conductivity ($\kappa$) along the *b*-axis direction of 11 TiS$_3$ nanoribbons with different thicknesses (Table 1) in the temperature range from 20 to 300 K, which displays several interesting trends. First, for all samples, $\kappa$ displays consistent temperature dependence, with its value first increasing with temperature from 20 to ~85 K and then decreasing as the temperature further escalates. Second, $\kappa$ shows a clear and unexpected thickness dependence for thinner ribbons (<52-nm thick). For example, at room temperature, $\kappa$ drops from ~15.5 W/m·K for the 19 nm thick ribbon to the bulk value of ~2.1 W/m·K for ribbons of >80-nm thick (Fig. 2b), which is opposite to the expectation based on the classical size effect as reported in a recent first-principles study[28]. To explore width dependence, we measured three samples with different widths, all of 52-nm thick, and the results show an increasing trend with the sample width, as shown in Supplementary Fig. 2. To explore the length dependence, we measured a 26-nm-thick ribbon with different suspended lengths (see Supplementary Note 4), and Fig. 2c indicates that the room temperature thermal conductivity increases from 7.3 to 9.2 W/m·K as the suspended length increases from 4.6 to 6.6 μm, but only varies marginally (<1.5%) as the length further extends from 6.6 to 8.9 μm. Overall, while the positive correlation with sample width or

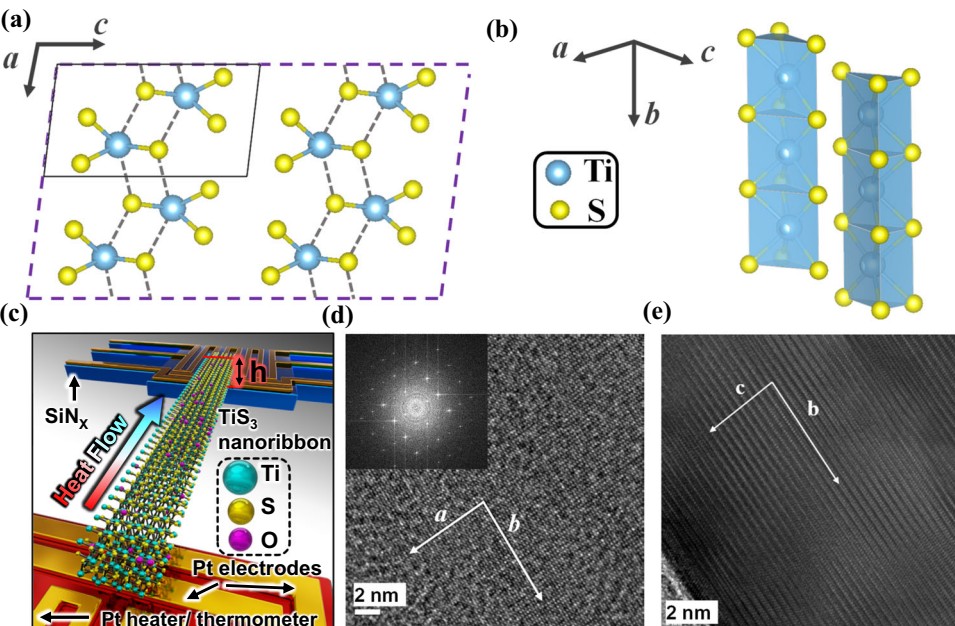

**Fig. 1 | Structure and characteristics of TiS$_3$.** Crystalline structure of TiS$_3$ within *a*–*c* plane (**a**) and along *b*-axis (**b**). The solid lines in (**a**) outline the unit cell, which has two Ti atoms and six S atoms. **c** A schematic diagram of a nanoribbon bridging two suspended membranes in a thermal measurement device. HRTEM micrographs viewed along [001] (**d**) and [100] (**e**) directions of a measured TiS$_3$ nanoribbon. The inset in (**d**) shows a fast Fourier transform pattern of an HRTEM image.

**Table 1 | The geometrical parameters of TiS$_3$ samples for thermal measurement**

| Sample no. | Thickness (nm) | Width (nm) | Suspended length (μm) |
|---|---|---|---|
| 1 | 19 | 310 | 4.74 |
| 2 | 26 | 98 | 6.57 |
| 3 | 30 | 148 | 8.67 |
| 4 | 34 | 126 | 4.23 |
| 5 | 43 | 152 | 8.60 |
| 6 | 52 | 65 | 6.38 |
| 7 | 52 | 132 | 6.69 |
| 8 | 52 | 220 | 10.20 |
| 9 | 62 | 231 | 10.26 |
| 10 | 80 | 190 | 5.27 |
| 11 | 170 | 160 | 6.61 |
| 12 | 256 | 300 | 6.71 |
| 13 | 272 | 274 | 6.85 |

The thickness is measured using an atomic force microscope, while the length and width are measured with a scanning electron microscope.

**Table 2 | Lattice constants of 20 TiS$_3$ samples obtained from electron diffraction pattern**

| Sample no. | Thickness (nm) | Width (nm) | a (nm) | b (nm) | Lattice contraction (%) |
|---|---|---|---|---|---|
| S1 | 19 | 310 | 0.4823 | 0.3179 | 9.3 |
| S2 | 25 | 243 | 0.4746 | 0.3282 | 7.8 |
| S3 | 26 | 190 | 0.4844 | 0.3277 | 6.1 |
| S4 | 26 | 211 | 0.4749 | 0.3365 | 5.4 |
| S5 | 33 | 233 | 0.4758 | 0.3300 | 7.1 |
| S6 | 34 | 213 | 0.4795 | 0.3286 | 6.8 |
| S7 | 35 | 242 | 0.4807 | 0.3302 | 6.1 |
| S8 | 42 | 190 | 0.4828 | 0.3297 | 5.8 |
| S9 | 52 | 220 | 0.4685 | 0.3382 | 6.2 |
| S10 | 57 | 161 | 0.4855 | 0.3347 | 3.8 |
| S11 | 62 | 231 | 0.4711 | 0.3384 | 5.7 |
| S12 | 79 | 298 | 0.4846 | 0.3397 | 2.6 |
| S13 | 90 | 251 | 0.4938 | 0.3362 | 1.8 |
| S14 | 105 | 334 | 0.4955 | 0.3373 | 1.1 |
| S15 | 108 | 367 | 0.4946 | 0.3366 | 1.5 |
| S16 | 110 | 320 | 0.4957 | 0.3365 | 1.3 |
| S17 | 127 | 436 | 0.4944 | 0.3367 | 1.5 |
| S18 | 137 | 381 | 0.4947 | 0.3365 | 1.5 |
| S19 | 182 | 466 | 0.4949 | 0.3385 | 0.9 |
| S20 | 272 | 274 | 0.4974 | 0.3389 | 0.3 |

length agrees with the classical size effect, the negative correlation with sample thickness is counterintuitive.

To understand the thickness dependence of thermal conductivity, we first evaluated the electronic contribution to $\kappa$ based on the Wiedemann–Franz law, $\kappa_e = LT\sigma$, where $L$ is the Lorenz number, $T$ is the temperature and $\sigma$ is the electrical conductivity. The measured $\sigma$ (Supplementary Fig. 3) corresponds to a rather low $\kappa_e$ that contributes <0.02% to $\kappa$ for all samples, thus phonons dominate $\kappa$. According to the kinetic theory, the lattice $\kappa$ is proportional to $Cv^2\tau$, where $C$ is the volumetric heat capacity, $v$ is the speed of sound and $\tau$ is the relaxation time. In a recent publication on thermal transport through NbSe$_3$ nanowires[29], it has been shown that elastic stiffening leads to a 25-fold increase of $\kappa$ as the wire hydraulic diameter reduces from 26 to 6.8 nm.

It has also been shown that elastic stiffening can enhance the lattice $\kappa$ of silver nanowires[30]. As such, we measured Young's modulus ($E$) of TiS$_3$ nanoribbons and examined whether elastic stiffening occurs in thin ribbons here.

### Young's modulus of TiS$_3$ nanoribbon

$E$ of TiS$_3$ nanoribbons along the $b$-axis was measured with a three-point bending scheme (Fig. 2d) using an atomic force microscope (AFM) for nine different samples with thickness spanning from 23 to 218 nm (Supplementary Table 1). A typical force–deflection (F–D) curve is plotted in Fig. 2f, from which $E$ is extracted (Supplementary Note 5). Consistent with the thermal conductivity data, $E$ remains approximately a constant value for ribbons thicker than ~80 nm but escalates rapidly as the ribbon thickness reduces below 50 nm as shown in Fig. 2e. The correspondence between $\kappa$ and $E$ in terms of both thickness dependence and transition thickness strongly suggests that the unexpected thickness dependence of $\kappa$ is induced by elastic stiffening with enhanced $E$ for thinner ribbons.

Elastic stiffening in thin nanowires has been observed for a wide variety of different wires. In addition to NbSe$_3$ and Ag nanowires mentioned previously, it has been shown that Au, CuO, Pb and ZnO nanowires also demonstrate elastic stiffening[31–34]. The reason for the increased $E$ is usually attributed to surface effects such as surface tension or enhanced surface elastic stiffness[35]. We note that for all 3D materials, the observed enhancement in $E$ is smaller than two-fold, while for TiS$_3$ nanoribbons, we observed a more than six-fold increase, similar to the case of NbSe$_3$ nanowires, which are also composed of quasi-1D vdW crystal. Therefore, it is likely that the enhancement of $E$ in quasi-1D vdW crystal nanowires could be due to a different mechanism.

### Lattice contraction due to atomic substitution

From a microscopic point of view, Young's modulus can be regarded as the ratio of interatomic spring constant to equilibrium lattice constant. Thus, we measured the thickness dependence of the lattice constant along the $a$- and $b$-axis directions from HRTEM images (Fig. 3a–c). The measured data are listed in Table 2, which suggests a significant reduction of lattice constant, i.e., lattice contraction, for thinner ribbons. To quantitatively measure the level of lattice contraction, we define a parameter of lattice contraction rate as $(0.169 - a*b)/0.169$, where 0.169 is the product of equilibrium lattice constants $a$ and $b$ for bulk TiS$_3$[27]. Figure 3d plots the thickness-dependent lattice contraction rate, which indicates that significant lattice contraction occurs for ribbons with thickness below 50 nm. The transition thickness agrees with the trend observed for $\kappa$ and $E$. We note that the lattice contraction rate can reach up to ~9% for a 19-nm-thick nanoribbon. Since lattice contraction shortens the interatomic interaction distance and significantly strengthens the interactions, Young's modulus of thin TiS$_3$ nanoribbon has a remarkable enhancement.

To evaluate the effects of lattice contraction, we calculated $E$ by first-principles calculations based on the experimental lattice contraction rates (see Methods). Encouragingly, the calculated values (triangle points) in Fig. 4a agree with the measurement data (rectangle points) in terms of both the trend and level of enhancement. In addition, we also used simple two-body potentials to examine the lattice contraction effects on Young's modulus (More details in Supplementary Note 9). Results (Supplementary Figs. 6 and 7) show that a 9% lattice contraction rate can induce a drastic enhancement of interatomic interaction strength and Young's modulus, which corroborates the first-principles calculations and experimental measurements (Fig. 4a). Thus, we conclude that the unexpected thickness dependence of $\kappa$ is caused by the enhancement in $E$, which is induced by the lattice contraction. In fact, $\kappa$ and $E$ show impressive agreement as the lattice contraction rate escalates as shown in Fig. 4a. Naturally, the key question is what leads to the lattice contraction.

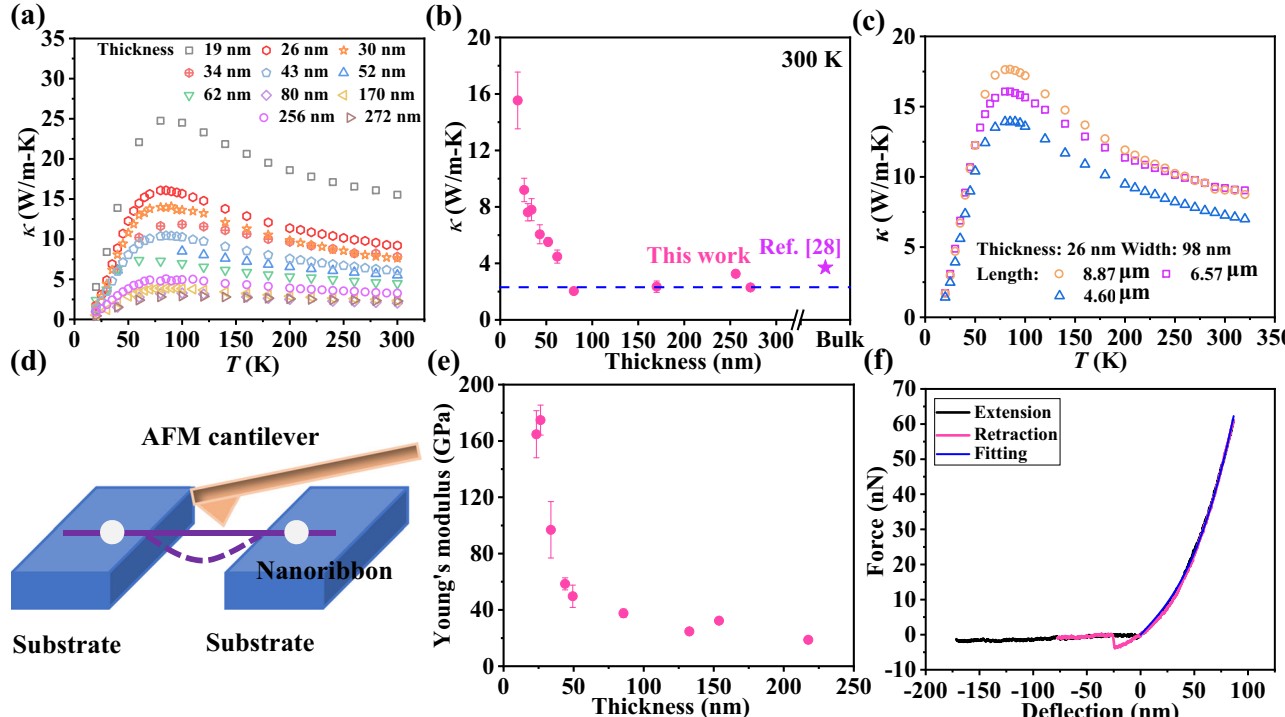

**Fig. 2 | Thermal and mechanical properties of TiS₃ nanoribbons.** The measured thermal conductivities $\kappa$ of TiS₃ nanoribbons along *b*-axis versus temperature $T$ for samples with different thicknesses (**a**) and lengths (**c**). The thickness-dependent room temperature thermal conductivity (**b**) and Young's modulus (**e**). The star in (**b**) means the bulk thermal conductivity from the previous report[27]. **d** Schematic diagram of three-point bending scheme for the measurement of Young's modulus. **f** The measured force–deflection (F–D) curve during the extension and retraction

phase. Young's modulus can be extracted by fitting the F–D curve. The error bars in thermal conductivity represent uncertainties evaluated based on measurement errors in thermal conductance, nanowire cross-section and length (see Supplementary Note 6). The error bars in Young's modulus represent uncertainties calculated based on measurement errors in the AFM cantilever spring constant, nanowire cross-section and length determined as the standard deviation from three individual measurements.

Based on Vegard's law[36,37], lattice contraction typically happens when host atoms are substituted by dopant atoms with smaller radii. Considering that materials can be oxidized easily during growth or in the ambient environment, it is likely that O atoms may substitute S atoms in thin TiS₃ nanoribbons, which are responsible for the observed lattice contraction. To verify this hypothesis, we measured the O atom concentration in TiS₃ nanoribbon samples by energy dispersive spectroscopy (EDS). As shown in Fig. 4b, the O atom concentration increases significantly as the ribbon thickness reduces. We note that although the measured O concentration with EDS may associate with large uncertainties[38], the systematic trend and the dramatic percentage increase as the ribbon thickness reduces provide convincing evidence that the O atom concentration increases in thinner ribbons. Moreover, as displayed in the inset of Fig. 4b, the increased peak intensities corresponding to the O atoms in the EDS spectrum also certify the increased O concentration with decreasing thickness. The consistent trends between the thickness-dependent lattice contraction rate and oxygen concentration suggest that the amount of O atoms in the ribbon could play a key role in the observed lattice contraction.

Although the existence of O atoms has been verified in the TiS₃ ribbon, it is not clear whether the O atoms are only adsorbed on the surface or they diffuse inside and substitute S atoms in the ribbon. In order to clarify this, we performed first-principles calculations to explore the relation between the lattice contraction rate and oxygen concentration. For the calculation, a portion of S atoms in the unit cell were directly substituted by O atoms and the symmetry constraint was switched off during the structural relaxation process. The calculated lattice contraction rate matches well with the experimental data (Fig. 4c) for different O concentrations. For comparison, $E$ of a lattice with O atoms adsorbed on the surface was calculated and the value is

even smaller than that of bulk TiS₃ (Supplementary Fig. 8). These results indicate that O atoms substitute S atoms in our samples.

Since defects can also cause lattice contraction in TiS₃ samples, we calculated the total energy and lattice contraction rate for TiS₃ with S vacancies. The results indeed show a clear lattice contraction with S vacancies; however, the total energy in TiS₃ with S vacancies is 7–17% higher than that of pristine TiS₃, while the corresponding value in the O substituted TiS₃ is 8.4% lower (Supplementary Table 2), indicating that the substitution of S atoms by O atoms is energetically favorable. To further rule out the possibility that our observation is induced by S vacancies, we calculated the phonon dispersion relations of pristine TiS₃, TiS₃ with S vacancies and TiS₃ with O substitution as shown in Supplementary Fig. 10. From these phonon dispersion relations, pristine TiS₃ and O substituted TiS₃ are stable while TiS₃ with one S atom vacancy is not stable because imaginary frequency phonons appear as shown in Supplementary Fig. 10a, b. Although the structure with two S vacancies (Supplementary Fig. 10c) is stable, the calculated thermal conductivity is actually lower than that of pristine TiS₃ along the *b*-axis direction (Supplementary Fig. 11), as a result of phonon-defect scattering. Moreover, the lattice with two S vacancies is quite different from that of TiS₃ and this structure is not consistent with our TEM study. Overall, while S vacancies can also cause lattice contraction, it will decrease the thermal conductivity and/or render the structure unstable. Therefore, we believe that the lattice contraction observed in our study is induced by the substitution of S atoms with O atoms. We note that a similar phenomenon was observed by Ning et al.[39], in which the volume of unit cell is reduced by 3.4% because of the intercalation of protons into WO₃ film.

To further verify the O atoms substitution, we measured the X-ray photoelectron spectroscopy (XPS) of a powder of TiS₃ nanoribbons.

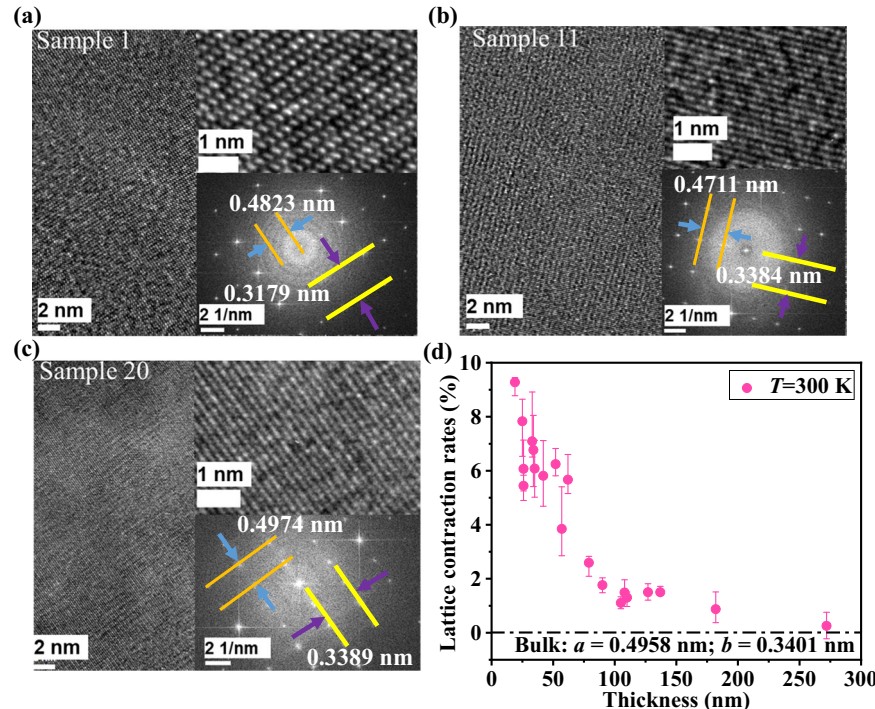

**Fig. 3 | Thickness-dependent lattice contraction. a** The HRTEM image of sample 1 (Table 2) with a ~30% O atom concentration. The upper right inset is a zoom-in view and the lower inset is a fast Fourier transform pattern. The HRTEM images of sample 11 (**b**) and sample 20 (**c**) are displayed for comparison. **d** Lattice contraction rates as a function of nanoribbon thickness measured from HRTEM at 300 K. The dot-and-dash line represents the situation of no lattice contraction from bulk TiS₃, where lattice constant $a$ is 0.4958 nm and $b$ is 0.3401 nm. The error bars of lattice contraction rates are from the variations among several individual measurements (see Supplementary Note 6).

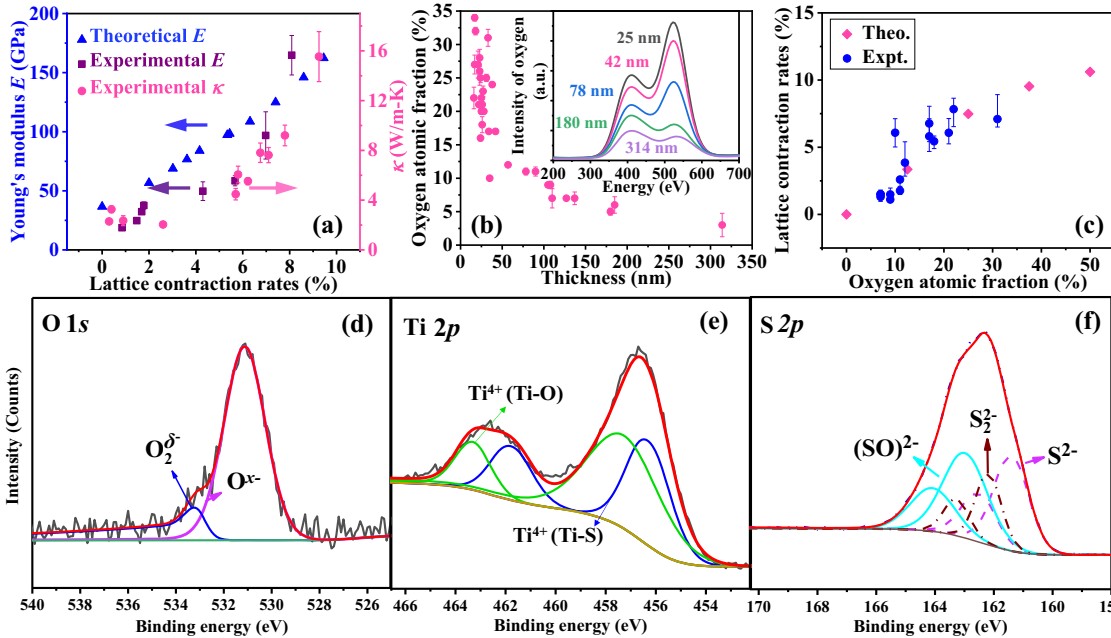

**Fig. 4 | Lattice contraction due to the substitution of S atoms by O atoms. a** The variations of Young's modulus $E$ with the lattice contraction rate. The rectangle points represent the data measured by AFM, while the triangle points are predicted from the first-principles calculations. The measured thermal conductivity data are also plotted with circle points while their values are labeled in the right vertical axis. The error bars in thermal conductivity represent uncertainties evaluated based on measurement errors in thermal conductance, nanowire cross-section and length (see Supplementary Note 6). The error bars in Young's modulus represent uncertainties calculated based on measurement errors in the AFM cantilever spring constant, nanowire cross-section and length determined as the standard deviation from three individual measurements. **b** The nanoribbon thickness-dependent O atom concentration measured with the EDS. The inset in (**b**) shows the intensity of O atoms for TiS₃ nanoribbons with different thicknesses. The error bars indicate the deviations from the integration of O peak obtained from EDS mapping. **c** The dependence of lattice contraction rate on O atom concentration from first-principles calculations and TEM measurements. The error bars are from the variations among several individual measurements (see Supplementary Note 6). **d**–**f** XPS spectra of O 1s, Ti 2p, and S 2p levels in O-doped TiS₃, respectively.

The atomic fraction of oxygen in the samples derived from the XPS spectrum is around 14%. While the XPS measurements cannot be performed on individual nanoribbons, the study further confirms that the ribbon contains O atoms. Figure 4d shows that the O 1s spectrum is composed of two peaks. The lower binding-energy peak at 531.11 eV is assigned to $O^{x-}$ $(0 < x < 2)$ and the higher binding-energy peak at 533.25 eV is characterized as $O_2^{\delta-}$ $(\delta < 2)$[40,41]. These less-negatively charged O species should be associated with a certain chemical bond formation[40]. The chemical state of $Ti^{4+}$ ions is further analyzed to confirm that the oxygen atoms substitute into the $TiS_6$ trigonal prisms. Figure 4e exhibits that the Ti 2p peak has been broken down into two components: the lower binding-energy peak (456.28 and 461.75 eV), which may be attributed to titanium atoms with a $TiS_6$ trigonal environment in $TiS_3$[42], and the higher binding-energy peak (457.27 and 463.31 eV), which has been found in titanium oxysulfide species and may be assigned to the Ti-O bond because the binding energy of the Ti-O bond is higher than that of Ti-S bond[43]. The S 2p peak containing three components is also investigated as shown in Fig. 4f. The lowest binding energy (161.36 and 162.46 eV) likely belongs to $S^{2-}$ sulfide ions in $TiS_3$ and the moderate binding energy (162.17 and 163.26 eV) could be attributed to $S_2^{2-}$ disulfide. In pristine $TiS_3$[44], the fitted disulfide/sulfide peak intensity ratio of S 2p spectrum is around 2, while our fitting results deviate from this value, indicating that other forms of S atoms exist. The highest binding energy (162.98 and 164.08 eV) is assigned to $(SO)^{2-}$ ions in trigonal prisms due to its shorter S-O bond. The detailed bond information is depicted in Supplementary Fig. 12. From the XPS analysis, a portion of S atoms are substituted by O atoms in thin $TiS_3$ nanoribbons.

### Comparison of doping effects in 3D and quasi-1D materials

Based on the above-detailed material characterizations and first-principles calculations, we can conclude that once the $TiS_3$ nanoribbon thickness decreases below 50 nm, more S atoms are substituted by O atoms, which leads to lattice contraction and enhanced thermal conductivity. As we know, doping is a widely adopted and effective strategy to reduce thermal conductivity in 3D materials, and the mechanism of the reduced thermal conductivity is attributed to the enhanced phonon-impurity scattering strength. It is curious that doping leads to an increase in the thermal conductivity in the quasi-1D vdW materials here. In order to reveal the underlying mechanism, the thermal conductivity of a bulk Si doped with C (inset of Fig. 5d) was

calculated. At a doping level of 25% ($Si_{0.75}C_{0.25}$), the lattice contraction appears due to the smaller radii of C atoms. However, the thermal conductivity decreases to 34% of the pure Si. Thus, it is unlikely that the increase of thermal conductivity in $TiS_3$ is only because of doping-induced lattice contraction, which should also originate from the quasi-1D vdW structure. The most distinct feature of quasi-1D vdW structure is the weak vdW interactions along the interchain directions and the presence of the vdW gaps. Then, we calculated the coupling strength along both the interchain and intrachain directions at different O doping levels (Fig. 5a). The coupling strength along the interchain directions fluctuates slightly, while the coupling strength along the intrachain direction demonstrates a significant increase with the O doping levels (Fig. 5b). Correspondingly, it is interesting to compare the strain fields induced by doped atoms in bulk silicon and $TiS_3$ nanoribbons. In bulk silicon, the doped C atom introduces lattice contraction along three-dimensional directions due to the strong covalent bonds of C-Si. As a result, the elastic strain induced by the doped C atoms in bulk Si is along all three-dimensional directions, and the strong potential energy and structural distortions are introduced, which significantly enhance phonon scattering and reduce phonon lifetime as demonstrated in Fig. 5d. In contrast, when O atoms substitute S atoms in $TiS_3$ nanoribbons, the vdW gaps provide the freedom and space for structural evolution. As a result, the doped atoms induce the lattice contraction and enhance the coupling strength only along the molecular chain direction (Fig. 5b), which leads to a significant increase in Young's modulus, phonon group velocity and phonon lifetime (Fig. 5c). Notably, the increased phonon lifetime is caused by the enlarged phonon bandgap due to lattice contraction (Supplementary Fig. 13). Overall, the difference of doping effects in quasi-1D and 3D materials causes the increase of thermal conductivity in O-doped $TiS_3$ and the decrease of thermal conductivity in C-doped Si.

Besides phonon group velocity and phonon lifetime, heat capacity is another factor to affect thermal conductivity. The lattice contraction increases the atom number density that will cause an increase in the heat capacity at room temperature. To prove it, we calculated the heat capacity through the first-principles method. Supplementary Fig. 14 verifies that heat capacity is enhanced at room temperature for the O-doped $TiS_3$ nanoribbon. Overall, Fig. 2e demonstrates that $E$ increases 6.1 times from 28 to 170 GPa, which corresponds to a 5.9-fold increase in the square of the speed of sound ($v^2 \propto E/\rho$). It is noticed

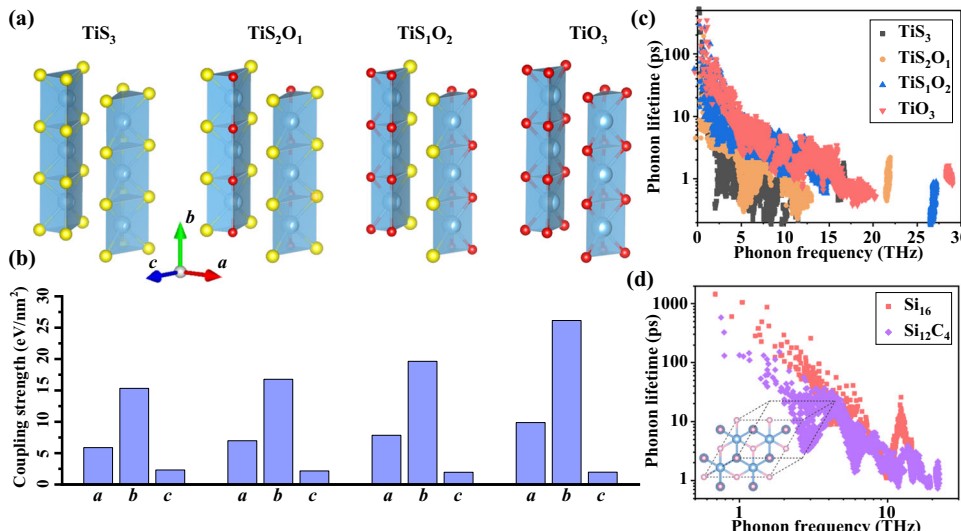

**Fig. 5 | Comparison of mechanical and phonon transport properties between quasi-1D TiS₃ doped with O atoms and bulk Si doped with C atoms. a** Atomic structures of $TiS_3$ at different O doping levels. **b** The coupling strength along the a-, b-, and c-axis directions in $TiS_3$. **c** Phonon frequency-dependent phonon lifetime due to three-phonon scatterings in pristine $TiS_3$ and O-doped $TiS_3$. **d** Phonon frequency-dependent phonon lifetime in pristine Si and C-doped Si.

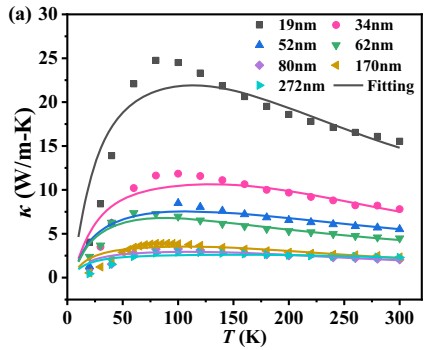
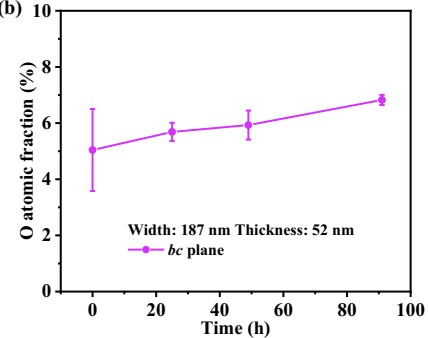

**Fig. 6 | Callaway model fitting and O atom concentration evolution with time. a** The Callaway model fitting of the measured thermal conductivities κ for TiS₃ nanoribbons with different thicknesses. **b** Time-dependent O concentration evolution for a 52-nm-thick TiS₃ by EDS measurement. The error bars indicate the deviations from the integration of O peak obtained from EDS mapping.

that the mass density $\rho$ increases from 3225.2 to 3317.2 kg/m³ as the thickness of TiS₃ nanoribbons reduces from ~80 to ~19 nm as a result of lattice contraction. Meanwhile, the thermal conductivity $\kappa$ increases by 7.4 times from ~2.1 to ~15.5 W/m-K at room temperature (Fig. 2b). This suggests that the enhancement of the speed of sound plays a key role in the increase of thermal conductivity. The specific heat and phonon lifetime have relatively weak contributions to this unexpected increase.

## Discussion

Considering that TiO₂ has a higher thermal conductivity than TiS₃, one might expect that the formation of a core-shell (TiS₃-TiO₂/TiOₓS₃₋ₓ) structure can explain the observed enhancement of thermal conductivity. However, detailed calculations exclude the possibility of the core-shell structure, which is clearly described in Supplementary Note 11. In fact, the diffraction pattern of HRTEM displays only one type of lattice structure. If a core-shell structure was formed, two types of lattice structures would be easily discerned because the weak interchain vdW interactions should allow for the TiS₃ and TiO₂/TiOₓS₃₋ₓ layers retaining their respective lattice structures.

We also note that the O-doping-induced lattice contraction is mainly thickness dependence, but not width and length dependence. This is because the coupling strength along the $a$-axis direction is 2.5 times higher than that along the $c$-axis direction as shown in Fig. 5b[28]. Thus, the sample width is always significantly larger than the thickness. Actually, the minimum width for all measured samples in the current work is >50 nm. As demonstrated in Fig. 4b, the thinner ribbons have a higher doping level because it is easier for O atoms to diffuse and replace S atoms. The O doping level starts to increase quickly as the ribbon thickness is below 50 nm. In contrast, the doping levels (and hence lattice contraction as shown in Supplementary Fig. 15a and Young's modulus) for all of the measured samples are not sensitive to

the width variations because the minimum width is larger than 50 nm. So, the measured thermal conductivity demonstrates strong thickness dependence but relatively weak width dependence. The relatively weak width-dependent thermal conductivity is attributed to the reduced phonon boundary scattering.

Atomic substitution causes mass mismatch and local strain, which inevitably enhances phonon-impurity scattering that tends to decrease thermal conductivity. However, our analysis suggests a longer phonon lifetime when O atoms replace S atoms in TiS₃ nanoribbon, which indicates that the reduction in the Umklapp scattering rate is more significant than the enhancement in phonon-impurity scattering rate. In order to further understand the effects of phonon-impurity scattering from O substitution, we applied the Callaway model to fit the temperature-dependent thermal conductivity and extracted the phonon-impurity scattering cross-section $\Omega$ and strength $A$ (where phonon-impurity scattering is expressed as $\tau_d^{-1} = A\omega^4 = \frac{d^3\Omega}{4\pi\nu_g^3}\omega^4$ based on Abeles' theory[45], and the detailed method was described in Supplementary Note. 7). Notably, the bulk phonon spectrum is employed in the model to fit the thermal conductivity since the thinnest sample (~19 nm) in the current study is larger than 20 atomic layers thick (17.6 nm)[28]. The fitted profiles are shown in Fig. 6a. The extracted scattering cross-section $\Omega$ increases monotonically with the decrease of nanoribbon thickness as listed in Table 3, which is reasonable because $\Omega$ describes the effects of mass disorder and local strain on phonon transport[39]. Interestingly, although the scattering cross-section increases significantly, the scattering strength $A$ in Table 3 has a weak dependence on thickness and even decreases as the thickness becomes smaller than 52 nm. The reason behind the decrease of $A$ is the significant increase in phonon group velocity $\nu_g$ (Supplementary Eq. 16) due to the lattice contraction. A similar decreasing trend of $A$ due to the increase of $\nu_g$ is observed in epitaxial WO₃ thin films[39]. Another possible reason could be that the dopant O atoms are located in the same group of the periodic table as S atoms. Thus, the two elements have the same number of electrons in the outermost electron shell, and subsequently, the phonon-impurity scattering due to O doping is suppressed. In order to verify our hypothesis, F atoms were chosen to substitute S atoms and the first-principles calculations observed much stronger phonon-phonon scattering than pristine TiS₃ and the O atom-doped TiS₃ (Supplementary Fig. 16). By employing Callaway model to fit the temperature-dependent thermal conductivity of F-doped TiS₃ from first-principles calculations (Supplementary Fig. 17), we find that both $\Omega$ and $A$ of F-doped TiS₃ are much higher than that of pristine TiS₃. Thus, two rules should be satisfied in order to increase thermal conductivity by means of doping: (1) dopant atoms should have a smaller radius compared with host atoms; (2) dopant atoms should locate in the same group of the periodic table as host atoms.

**Table 3 | The phonon-impurity scattering cross-section $\Omega$ and strength $A$ as well as the lattice contraction rate and thermal conductivity at room temperature**

| Sample no. | Ω (a.u.) | A (a.u.) | Lattice contraction rate (%) | κ (W/m-K) |
|---|---|---|---|---|
| 1 | 36,489 | 4.57 | 9.3 | 15.54 |
| 4 | 15,938 | 5.07 | 6.8 | 7.81 |
| 8 | 7856 | 5.13 | 6.2 | 5.52 |
| 9 | 5796 | 4.75 | 5.7 | 4.48 |
| 10 | 9980 | 12.30 | 2.6 | 2.04 |
| 11 | 4777 | 7.97 | 0.9 | 2.35 |
| 13 | 2388 | 8.21 | 0.3 | 2.29 |

The sample numbers here correspond to those in Table 1.

Finally, we want to briefly discuss how to intentionally control the oxygen doping level in TiS$_3$ since the doping level in the current study is controlled by nanoribbon thickness, which is not intentional. As discussed above, the atomic modeling indicates that the O atom-doped TiS$_3$ has a lower energy compared to pristine TiS$_3$, indicating that the substitution of S by O atoms is energetically favorable. In order to verify the modeling result, the evolution of O atom concentration with the elapse of time was experimentally measured and the details are shown in Supplementary Fig. 19, which demonstrates that O atoms could adsorb and then replace S atoms in TiS$_3$ samples though the substitution rate is rather slow in ambient conditions as shown in Fig. 6b. Thus, the substitution of S by O atoms is spontaneous in TiS$_3$, which is similar to that the O atoms can replace S atoms in monolayer MoS$_2$ under ambient conditions[46]. If the sample is put into an oxygen-rich environment, the substitution rate should be high, which may provide an intentional method to control the oxygen doping level.

In this work, we explore thermal transport in quasi-1D vdW crystal TiS$_3$ nanoribbons. The measured thermal conductivity displays an unexpected increase as the ribbon thickness reduces, which is opposite to the expectation based on the classical size effect on phonon transport. Measurements of the corresponding mechanical properties indicate that this thermal behavior is due to elastic stiffening with a drastic increase of the ribbon Young's modulus of up to six times the bulk value. Detailed structural characterizations and composition analyses show that the unexpected thermal and mechanical properties are induced by lattice contraction as a result of the substitution of a portion of sulfur atoms with oxygen atoms. Due to the presence of the vdW gaps in TiS$_3$, the lattice contraction enhances the bond strength significantly along the molecular chain direction while weakly affects vdW strengths, leading to the enhancement of thermal conductivity. These findings provide new insights into the structural properties of quasi-1D vdW nanostructures and potential approaches of modulating their mechanical and thermal properties.

## Methods

### TiS$_3$ nanoribbon fabrication
TiS$_3$ nanoribbons were grown in a furnace with titanium and sulfur powders loaded according to the stoichiometric ratio. The reactants were placed in a quartz ampule, evacuated to <1 Pa and flame-sealed. The ampule was then heated to 500 °C within two hours and held at this temperature for 2.5 days in a box furnace for a complete reaction via a chemical vapor transport process. As a result, TiS$_3$ nanoribbons were grown and distributed around the inner wall of the quartz ampule, as reported in the literature[47].

### Structural characterizations
TEM and high-resolution TEM studies were carried out to measure the lattice constant of TiS$_3$ nanoribbon via using Titan 80-300, FEI microscope. The thickness of samples was measured by AFM (Cypher S, Asylum Research, Oxford Instruments). The width and length of specimens were obtained from scanning electron microscopy (Helios NanoLab 600i, FEI, USA) images. The XPS measurement was performed by Al Kα source that provides monochromatic x-rays at 1486.6 eV. Narrow-scan spectra of concerned regions were recorded to analyze the chemical binding states of corresponding elements. The obtained binding energies were calibrated with the contaminant C$_{1s}$ peak at 284.6 eV. The narrow-scan spectra were fitted through a Gaussian−Lorentzian product function.

### Thermal conductivity, electrical conductivity and Young's modulus measurements
The thermal resistance and electrical conductance of samples were measured using the suspended micro-devices technique following previous reports[48–50]. The suspended micro-thermometry devices as illustrated in Fig. 1c include two adjacent silicon nitride (SiN$_x$) membranes suspended with six SiN$_x$ beams severing as heating source and heating sink, respectively. The samples were measured in a cryostat under a high vacuum in order to eliminate the heat transfer through the air. The traditional 4-point I-V measurement approach was implemented to investigate the electrical transport in TiS$_3$ NRs[30,51].

To measure Young's modulus, individual TiS$_3$ NRs were transferred to a 6-μm-wide Si trench using micromanipulator. The Pt powder was deposited at the two ends of the nanoribbon to clamp the ribbon to the substrate. Then we carried out the three-point bending test with AFM. Before bending test, the spring constant of cantilever was extracted via thermal tune process. Figure 2d shows a schematic description of the bending experiment. First, we carefully imaged the devices using the AFM to locate the middle point across the trench and then pushed the middle point of the nanoribbon using the AFM cantilever to implement bending test. The bending tests were repeated three times to guarantee the rigidity of the constraints and the elastic property of the nanoribbons within the measured range.

### First-principles calculations of Young's modulus
To calculate Young's modulus, a large energy cutoff of 600 eV was chosen with the Perdew−Burke−Ernzerhof of generalized gradient approximation as the exchange-correlation functional[52]. The Γ-centered **k**-mesh was set as $15 \times 25 \times 9$ to simulate the Brillouin zone in the total energy calculation and structural optimization. To introduce the vdW interactions into the TiS$_3$ nanostructure, we imposed the density functional dispersion correction method of Grimme (DFT-D3)[53] in all calculations. The tolerances for geometry optimization were set as the difference in atomic force being within $10^{-3}$ eV/Å. In calculating Young's modulus for O-doped TiS$_3$, lattice constants were adjusted according to experimental lattice contraction rates. The steps to calculate Young's modulus are (1) the elastic tensor is determined by performing six finite distortions of the lattice; (2) the elastic constants are obtained based on the previous elastic tensor and the strain−stress relationship[54]; and (3) Young's modulus of TiS$_3$ were derived from elastic constants metrics through a series of formulas given in ref. 55.

### Theoretical calculations of specific heat and phonon lifetime of O-doped TiS$_3$
To investigate the doping effects of O atoms on phonon thermal transport quantitatively, thermal conductivity of TiS$_3$ with different O concentrations was calculated with the first-principles method. In the calculation, energy cutoff of 520 eV was chosen and DFT-D3 method was employed to consider the vdW interactions. The Γ-centered **k**-mesh was set as $15 \times 25 \times 9$ for bulk TiS$_3$ with different O atom concentrations to simulate the Brillouin zone in the total energy calculation and structural optimization. The lattice constants and internal atomic coordinates were optimized until the atomic forces became less than $10^{-3}$ eV/Å. To calculate the harmonic interatomic force constants, we chose the $3 \times 4 \times 2$ supercell with Γ-centered $3 \times 3 \times 2$ **k**-mesh. To obtain the anharmonic interatomic force constants, a $3 \times 4 \times 2$ supercell with Γ-centered $2 \times 2 \times 2$ **k**-mesh was chosen and the cutoff distance was set as 0.41 nm. The specific heat and phonon lifetime were calculated in the ShengBTE code[56] by iteratively solving the linearized BTE.

## Data availability
The data generated in this study have been deposited in Figshare under accession code https://doi.org/10.6084/m9.figshare.24039186.

## Code availability
The code that has been used for this work is available from the corresponding author on request.

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

## Acknowledgements

The authors thank the financial support from the National Natural Science Foundation of China (No. 52127811, 52035003, 52206092) and the Department of Science and Technology of Jiangsu Province (BK20220032). C.L. was funded by the Natural Science Foundation of Jiangsu Province (Grant no. BK20210565), Basic Science (Natural Science) Research Project of Higher Education Institutions of Jiangsu Province (21KJB470009), "Shuangchuang" Doctor program of Jiangsu Province (JSSCBS20210315), and the open research fund of Jiangsu Key Laboratory for Design and Manufacture of Micro-Nano Biomedical Instruments, Southeast University (No. KF202010). The numerical calculation in this research work is supported by the Big Data Computing Center of Southeast University and the Scientific Computing Center of Nanjing Normal University.

## Author contributions

C.L. performed theoretical calculations and wrote the manuscript with the assistance of C.W. X.Y.T. fabricated samples. C.W. performed thermal measurements and sample characterizations. C.L., D.L., and Y.C. analyzed the results. C.W., Y.T., and Y.Z. discussed the results. Y.C., Q.Y., and J.Y. conceived and directed the project.

## Competing interests
The authors declare no competing interests.
