## [Peer Review File · Nature Communications]

Unexpected doping effects on phonon transport in quasi-one-dimensional van der Waals crystal TiS₃ nanoribbonsREVIEWER COMMENTS

Reviewer #1 (Remarks to the Author):

The authors explored the unexpected enhanced thermal conductivity of TiS₃ when the thickness decreases. The various material characterizations and first-principles calculations have been utilized to explaining the reasons for the enhancement of thermal conductivity with decreasing thickness. The possible mechanism behind this phenomenon is interesting. The following issues should be addressed before its reconsideration:

1. The thickness dependent enhancement of thermal conductivity of TiS₃ was already reported in ref. [26]. Therefore, The innovation point of this paper needs to be more explicit.
2. The TiS₃ samples seem to possess S defect easily. Will the S defect also contribute to the lattice contraction?
3. The samples seem to be randomly selected to conduct a serial of measurement and these samples have different fraction of O atoms. When does the O atoms replace the S sites? Does it occur when the natural process of crystal growth or crystal exfoliates? Is it possible to conduct the measurement in the same sample as a reference to the measurement in the paper? For example, when measuring Young's modulus in the one thickness condition, could the authors exfoliate some layers and then measure the Young's modulus?
4. The English should be improved, for example, "first, the enhanced E will leads to a higher speed of sound as".

Reviewer #2 (Remarks to the Author):

Wu et al report size and doping effect of TiS₃ nanoribbons and find an increasement of thermal conductivity with decreasing thickness. Authors attribute this "abnormal behavior" to the Oxygen doping introduced lattice contraction and the corresponding increasement of Young's modulus. I do not fully agree with author's comment of the size-confinement thermal conductivity since this result is simply due to the oxygen doping (see details below). Also, it looks that 30% S is replaced by O (figure 4b), then what's exactly the authors measured, TiS₃ or TiO? Few more comments are listed:

1. I do believe that thermal conductivity increase with O doping since TiO₂ have much higher thermal conductivity. O doping should happen during cleavage process. This doping should only affect few nanometers near the sample boundary. Therefore, the total thermal conductivity should consist of three parts: the outer TiOS₃ and in the inner TiS₃, with the latter have smaller thermal conductivity. With sample thickness decreases, the outer TiOS₃ layer become more dominate and therefore thermal conductivity increases. Therefore, the measured result does not related to the size confinement as author claimed in the Abstract. This physical picture can also explain my second comment. The real size confinement/dependent thermal conductivity of TiS₃ has already be done by reference 21-23 as author wrote from line 58 to line 59.
2. TiS₃ is a quasi-1D system (can be reflected by the near linear $k \sim T$ at low temperature), meaning a-axis and c-axis are isotropic with van der Waals interaction. Then why thermal conductivity increases with width but decrease with thickness? This is related to my comment 1.
3. Figure 2a is also beyond my understanding. Thermal conductivity should increase with temperature as T^3 for 3D systems and nearly linear for 1D. However, for example, the thermal conductivity for sample with 272nm-thickness is already approaching zero, far

beyond the linear k - T curve. The thermal conductivity will be negative when temperature approaching zero.

4. According to $v=(E/\rho)^{0.5}$, the speed of sound should also related to mass density. Since S is replace by O, and introduces lattice contraction, the mass density must change.

5. Thermal conductivity increasement when S doped by O is quite normal since TiO₂ has much higher thermal conductivity.

Reviewer #3 (Remarks to the Author):

Authors have presented an enhancement of the thermal conductivity of TiS₃ nanoribbons with a reduction of their thickness, and discussed its origin based on the elastic stiffening. Experimental results authors provided are of high quality and the main argument about the elastic stiffening is supported well experimentally and theoretically. However, I am not convinced that the present work delivers a message important enough to be shared in Nature Communications.

As authors cited as Ref. 24, a similar experimental observation was already reported for similar quasi-1D vdW NbSe₃ nanowires. In Ref. 24, the hydraulic diameter is varied in nanometer scales, and the thermal conductivity becomes enhanced by 25 times. Authors there also determined the Young's modulus and concluded that the elastic stiffening is responsible for the thermal conductivity enhancement. In the present work, authors made similar studies including the enhancements of both the thermal conductivity and Young's modulus in the quasi-1D vdW TiS₃ nanoribbons. I found that TEM and XPS measurements were additionally conducted to further characterize lattice parameters and chemical compositions. Although authors provided two rules about dopant atoms for the thermal conductivity enhancement, I am not sure that this can guarantee a novelty for the publication in Nature Communications.

Authors often mentioned that the thickness dependence of thermal conductivity is counterintuitive when considering the size effect and confinement effect. There should be a more detailed supporting discussion for this argument.

In line 163, authors compared the lattice contraction rate and Young's modulus enhancement quantitatively. There should be a further explanation why the lattice contraction rate is not enough to explain the observed enhancement of the Young's modulus.

In the discussion around line 311, how is the phonon-impurity scattering varies when F atoms are chosen as dopant atoms?

Overall, the experimental observations and the discussions have been presented quite logically. And, the conclusion is well supported. Nevertheless, the novelty of the present work does not seem to high enough, and therefore I recommend this paper would be shared in a more specialized journal.

Response to the Reviewers' Comments

We sincerely thank the reviewers for the valuable comments on our manuscript. We have carefully considered all comments and made revisions accordingly, as reflected in the point-by-point response below. We believe that the revisions have significantly improved the quality of the manuscript and hope the reviewers find the updated version suitable for publication.

Reviewer #1

General remark: *The authors explored the unexpected enhanced thermal conductivity of TiS₃ when the thickness decreases. The various material characterizations and first-principles calculations have been utilized to explaining the reasons for the enhancement of thermal conductivity with decreasing thickness. The possible mechanism behind this phenomenon is interesting. The following issues should be addressed before its reconsideration.*

Response: We thank the reviewer for the favorable comments on our work and have addressed the concerns raised as follows.

Comment #1. *The thickness dependent enhancement of thermal conductivity of TiS₃ was already reported in ref. [26]. Therefore, the innovation point of this paper needs to be more explicit.*

Response: We thank the reviewer for bringing this up. The main innovation point of this manuscript is how substitution of a portion of S atoms in TiS₃ with O atoms drastically reduces the lattice constant and enhances the Young's modulus and the thermal properties of TiS₃ nanoribbons. The percentage of O atoms increases as the ribbon thickness reduces, which leads to enhanced thermal conductivity. Our current work experimentally demonstrates that nanoribbon thickness can influence doping level, which can also be employed as a method to enhance lattice thermal conductivity. In contrast, Ref. [26] is a pure theoretical study suggesting a thickness dependent thermal conductivity of layered TiS₃, which is a direct result of size confinement on phonon scattering rates while the material remains as pure TiS₃. This difference in the underlying mechanism also leads to a different thickness dependence. In this manuscript, the thermal conductivity of TiS₃ nanoribbons starts to escalate as the thickness becomes thinner than ~80 nm and keeps the increasing trend to the thinnest ribbon of ~19 nm thick. In contrast, Ref. [26] suggests a non-monotonic thickness dependence with a minimum thermal conductivity at ~17.6 nm thick.

In response to the reviewer's comment, we have emphasized the innovation of this manuscript as compared to Ref. [26] as follows.

Lines 304-313 in page 17-18 "Recently, a non-monotonic thickness dependent thermal conductivity of layered TiS₃ was reported by first-principles calculations,²⁶ which suggests a minimum thermal conductivity at ~17.6 nm thick. In the current work, the thermal conductivity of TiS₃ nanoribbons starts to escalate as the thickness becomes thinner than ~80 nm and keeps the increasing trend to the thinnest measured ribbon of ~19 nm thick. We note that the non-monotonic thickness dependent thermal conductivity in Ref. 26 is for pristine TiS₃ as a result of phonon confinement purely due to size effect while the significantly enhanced thermal

conductivity with decreasing thickness in this study is caused by the O substitution induced lattice contraction. This difference in the underlying mechanism leads to the different thickness dependences in Ref. 26 and current study.”

Comment #2. *The TiS₃ samples seem to possess S defect easily. Will the S defect also contribute to the lattice contraction?*

Response: We thank the reviewer for this insightful comment. It is true that the S defect may contribute to lattice contraction and to explore this effect, we conducted additional first-principles calculations. In the modeling, one or two S atoms are removed from the unit cell of TiS₃ to mimic the S defects, as shown in Fig. R1. Note that in Fig. R1d, the vacancy is filled with O atom. The structure was then relaxed with the Γ -centered k -mesh of $11 \times 15 \times 7$ to simulate the Brillouin zone. To introduce the van der Waals (vdW) interaction into the TiS₃ nanostructure, we adopted the density functional dispersion correction method of Grimme (DFT-D3)¹ in all calculations. The tolerance for structure relaxation was set as the difference in atomic force being within 10^{-3} eV/Å.

Figure R1. Structure relaxation for TiS₃ with one S defect-1 (a), one S defect-2 (b), two S defects (c) and two O substitution (d). The yellow, blue and red balls in (a)-(d) represent S, Ti

and O atoms, respectively.

After structure relaxation, the lattice constant was extracted, and we found that the S defects could also cause lattice contraction (Table R1), as the referee suggested. However, compared with the total energy in one unit cell of pristine TiS_3 , the total energy in TiS_3 with S defects is 7%-17% higher while the total energy in the O substituted TiS_3 is 8.4% lower (Table R1). Thus, the substitution of S with O in TiS_3 is more stable and O atoms will spontaneously occupy the corresponding positions of S defects.

Table R1. Lattice constants, total energy per unit cell and lattice contraction rates of different structures as shown in Fig. R1.

Label	a (nm)	b (nm)	c (nm)	Total energy (eV)	Lattice contraction rates (%)
One S defect-1	0.4677	0.3346	0.8667	-43.39	12.7
One S defect-2	0.4869	0.3399	0.8476	-45.22	9.7
Two S defects	0.6303	0.3573	0.4859	-40.85	29.6
Two O substitution	0.4743	0.3280	0.8324	-52.48	16.6
Pristine TiS_3	0.4930	0.3427	0.9195	-48.39	0

Figure R2. Phonon dispersion relations of TiS_3 with one S defect-1 (a), one S defect-2 (b), two S defects (c) and two O substitution (d).

To further rule out the possibility that our observation is induced by S defect, we calculated the phonon dispersion relations of pristine TiS_3 , TiS_3 with S defects and O substitution as shown

in Fig. R2. From these dispersion relations, pristine TiS_3 and O substituted TiS_3 are stable while TiS_3 with one S atom defect is not stable due to the appearance of imaginary frequency as shown in Fig. R2a and b. Although the structure with two S defects (Fig. R2c) is stable, the calculated thermal conductivity is smaller than that of pristine TiS_3 along the b -axis due to the defect introduced phonon scattering, as shown in Fig. R3. Moreover, the lattice with two S defects is much different from that of pristine TiS_3 , which is not consistent with our TEM data. Overall, while S defects can cause lattice contraction, it will decrease the thermal conductivity and/or make the structure unstable. Therefore, we believe that the main mechanism for the lattice contraction observed in our study is induced by substitution of S atoms with O atoms.

Figure R3. Calculated thermal conductivity of pristine TiS_3 , TiS_3 with two S defects and two O substitution in unit cell.

To clarify this, we have edited the manuscript with a discussion about the effect of S defect as follow.

Lines 228-245 in page 14-15 “Through first-principles calculations, we also found the total energy of O substituted TiS_3 is lower than that of pristine TiS_3 , indicating that the substitution process is energetically favorable (Table S2). Since defects can also cause the lattice contraction in TiS_3 samples, we further calculated the total energy and lattice contraction of TiS_3 with S vacancies. The results indeed show a clear lattice contraction with S vacancies; however, the total energy in TiS_3 with S vacancies is 7%-17% higher than that of pristine TiS_3 , while the correspond value in the O substituted TiS_3 is 8.4% lower (Table S2). To further rule out the possibility that our observation is induced by S vacancies, we calculated the phonon dispersion relations of pristine TiS_3 , TiS_3 with S vacancies and O substitution as shown in Fig. S13. From these dispersion relations, pristine TiS_3 and O substituted TiS_3 are stable while TiS_3

with one S atom vacancy is not stable because imaginary frequency phonons appear as shown in Fig. S13 (a), (b). Although the structure with two S vacancies (Fig. S13 (c)) is stable, the calculated thermal conductivity is actually lower than that of pristine TiS_3 along the b-axis (Fig. S14), as a result of phonon-defect scattering. Moreover, the lattice with two S vacancies is quite different from that of TiS_3 and this structure is not consistent with our TEM study. Overall, while S vacancies can also cause lattice contraction, it will decrease the thermal conductivity and/or render the structure unstable. Therefore, we believe that the lattice contraction observed in our study is induced by the substitution of S atoms with O atoms.”

Comment #3. The samples seem to be randomly selected to conduct a serial of measurement and these samples have different fraction of O atoms. When does the O atoms replace the S sites? Does it occur when the natural process of crystal growth or crystal exfoliates? Is it possible to conduct the measurement in the same sample as a reference to the measurement in the paper? For example, when measuring Young’s modulus in the one thickness condition, could the authors exfoliate some layers and then measure the Young’s modulus?

Response: We thank the reviewer for this interesting question. In our studies, we find a systematic trend of increasing O concentration for thinner ribbons, which we believe leads to the observed lattice contraction, enhanced Young’s modulus and thermal conductivity. However, while we once briefly discussed about when the O atoms replace the S atoms, we never conducted experiments to clarify this. One clarification is that our nanoribbon samples are directly synthesized from the chemical vapor transport process, instead of prepared through exfoliation from a bulk sample.

Now in response to the reviewer’s question, we conducted a series of experiments, which suggests that O may replace S both in the growing process and after the growth once the sample is exposed to air, as disclosed by the studies described below.

First, right after the nanoribbons were taken out of the furnace, the samples were sealed in vacuum and transferred from Singapore to Nanjing, China. Few days later, X-ray photoelectron spectroscopy (XPS) was performed to quantify O atom concentration in the TiS_3 nanoribbons, which yielded an atomic fraction of oxygen of around 14%. This result indicates that the samples already contain O atoms before we conduct thermal and mechanical experiments.

In order to address the reviewer’s concern, we measured the averaged O atom concentration adsorbed in the bc plane as time elapses. Specifically, we exfoliated a 52 nm thick and 187 nm wide nanoribbon from a 238 nm wide and 538 nm thick sample to observe the evolution of the O atom concentration with the exposure time to air. The exfoliation and transfer processes were done in air for 4 hours. Right after we prepared the sample, Energy-dispersive X-ray spectroscopy (EDS) examination was performed on the exfoliated sample (52 nm thick). Fig. R4 shows the evolution of the O atom distribution in the bc -plane. From 0 to 25 h, the sample was kept in a vacuum chamber for 23 hours and then exposed to air for 2 hours. At this point, the EDS study clearly showed that the O atom concentrations on the two edges were higher than that in the center region of the bc plane. From 25 h to 49 h, the sample was placed in a vacuum chamber again and we measured the O concentration at the time point of 49 h, which indicated an enhanced and uniform O concentration across the bc plane. From 49 h to 91 h, the sample was kept in a vacuum chamber for 37 hours and placed in air for 5 hours. We observed

highly accumulated O atoms near the two edges again at 91 h. Then the O atomic fraction in the bc plane was measured at 0, 25, 49 and 91 h, respectively (Fig. R5).

Figure R4. The evolution of the O atom distribution across the bc plane of a TiS_3 nanoribbon with the elapse of time. The b -axis stands for the nanoribbon length direction while the c -axis stands for the ribbon thickness direction.

Although Fig. R4 suggests that O atoms can indeed be introduced through surface adsorption and diffusion process in air, the O atomic fraction shown in Fig. R5 only changes slightly, indicating that O atoms are easily adsorbed by the bc plane while only a very small portion of O atoms with thermal energy high enough are able to diffuse into the sample, which has been demonstrated in previous work². Overall, Figs. R4 and R5 tell us that O atoms could adsorb and then replace the S atoms in TiS_3 samples but the rate for O replacing S is very slow in ambient conditions. Thus, the O atoms were introduced during the crystal growth process in the current study.

In addition, we conducted a measurement with the Raman spectroscopy to verify the stability of TiS_3 nanoribbons, in which the samples were kept in vacuum for 3 years. Fig. R6 shows the four classical Raman-active modes of TiS_3 , which are consistent with previous works^{3, 4}, indicating the stability of TiS_3 lattice structure. In our experiments, all the samples were kept

in vacuum and the thermal and mechanical measurements were conducted right after the samples were prepared successfully.

Figure R5. The evolution of the density of O atoms in the *bc* plane with the elapse of time.

Lastly, we explain the difficulty to measure the Young's modulus of nanoribbons with varied thicknesses exfoliated from one initial sample. The Young's modulus of a nanoribbon is measured by the three-point bending test, in which the ribbon is laid over a trench with the two edges fixed and the middle point is pushed by an atomic force microscope (AFM) tip to apply a normal load to deform the nanoribbon, as shown in Fig. R7. Thin Pt films are locally deposited to weld the two ends of the nanoribbon on the substrate to avoid slip when pushed by the AFM tip. After the Young's modulus measurement, it is difficult to exfoliate a thinner ribbon from this fixed sample to conduct the next round measurement because the measured sample must be long enough to span over the trench with the two ends fixed. Another challenge comes from the requirement of a uniform cross section area along the nanoribbon axis direction. In order to satisfy this requirement, the measured sample is usually selected from examining quite a few samples. It is difficult to obtain an ideal sample from only one exfoliation process.

Figure R6. Raman spectra of TiS_3 nanoribbons with different size. The blue area denotes the Raman-active modes of Si substrate. The H and W represents thickness and width of nanoribbons, respectively.

Figure R7. An SEM micrograph showing a nanoribbon suspended over a trench on a silicon substrate for three-point bending test.

Comment #4. The English should be improved, for example, “first, the enhanced E will lead to a higher speed of sound as”.

Response: We thank the reviewer for pointing this out and have fixed issues related to the grammar.

Reviewer #2

General remark: Wu et al report size and doping effect of TiS_3 nanoribbons and find an increasement of thermal conductivity with decreasing thickness. Authors attribute this “abnormal behavior” to the Oxygen doping introduced lattice contraction and the corresponding increasement of Young’s modulus. I do not fully agree with author’s comment of the size-confinement thermal conductivity since this result is simply due to the oxygen doping (see details below). Also, it looks that 30% S is replaced by O (figure 4b), then what’s exactly the authors measured, TiS_3 or TiO ?

Response: We thank the reviewer for examining our manuscript and the valuable comments. It is true that up to 30% S is replaced by O for the 19 nm thick TiS_3 sample. We regard the O atoms as dopants because even with this high percentage of O atoms, the lattice structure of the nanoribbons remains the same as that of pristine TiS_3 but not TiO_2 , as disclosed by the HRTEM micrographs in the manuscript. Intuitively, the enhanced thermal conductivity could be attributed to the O doping because TiO_2 has a higher thermal conductivity than that of TiS_3 . However, the composition change itself cannot explain the significantly enhanced thermal conductivity, which we will discuss in more details in the response to comments 1 and 5.

We agree with the reviewer that the trend is not induced by size-confinement. The trend manifests a size/thickness dependence simply because it is easier for O atoms to diffuse through thinner samples, which leads to a higher doping level for thinner samples. We have further clarified this in the revised manuscript.

Comment #1. I do believe that thermal conductivity increases with O doping since TiO_2 have much higher thermal conductivity. O doping should happen during cleavage process. This doping should only affect few nanometers near the sample boundary. Therefore, the total thermal conductivity should consist of three parts: the outer TiOS_3 and in the inner TiS_3 , with the latter have smaller thermal conductivity. With sample thickness deceases, the outer TiOS_3 layer become more dominate and therefore thermal conductivity increases. Therefore, the measured result does not relate to the size confinement as author claimed in the Abstract. This physical picture can also explain my second comment. The real size confinement/dependent thermal conductivity of TiS_3 has already be done by reference 21-23 as author wrote from line 58 to line 59.

Response: First of all, the thermal conductivity of TiO_2 at room temperature has been experimentally measured and theoretically predicted with the values ranging from 1.3 to 15 W/m-K^{5, 6}, among which the values for nanoribbons are lower than the bulk value due to phonon-boundary scattering (Table R2). Importantly, even the highest value of TiO_2 is smaller than the ~15.5 W/m-K for the 19 nm thick nanoribbon sample in our measurement. In addition, the thermal conductivity of $\text{TiO}_x\text{S}_{3-x}$ should be less than TiO_2 , which would make it even more difficult to achieve the 15.5 W/m-K we measured. Note that we agree with the referee that our observation is not due to size confinement. The size effect mentioned in the abstract is related to the general trend that the thermal conductivity of nanostructures usually decreases as the characteristic size reduces, while we attribute the observed enhanced thermal conductivity to enhanced Young’s modulus that is caused by O doping induced lattice contraction.

Table R2. Thermal conductivity of TiO₂ at room temperature from experimental measurement and first-principles calculation.

type	Thickness or diameter	k (W/m-K)	References
film	150 nm	6.3-8.0	Expt. ⁶
	30 nm	4.6-5.3	
nanowire	250-400 nm	1.3-5.6 (Anatase phase)	Expt. ⁵
bulk	-	8.5 (Anatase phase)	
		10.0 (Rutile phase)	
		15.0 (Anatase phase)	

We believe the core-shell model that has been used to account for the elastic stiffening/softening effect of various nanostructures might not be the best picture for quasi-1D vdW nanostructures. The weak interchain vdW interactions should allow for easy diffusion of the O atoms to inner section of the nanoribbon and the distribution of the O atoms is rather uniform across the entire cross-section, as disclosed by the EDS study. In fact, the diffraction pattern of HRTEM displays only one type of lattice structure. If a core-shell structure is formed, there should exist two types of lattice structures, especially considering the significant lattice contraction. This is especially true given the weak interchain vdW interaction could allow for the TiS₃ and TiO₂/TiO_xS_{3-x} layers keeping their respective lattice structures.

Now we further show that it is unlikely that the core-shell model (Fig. R8a) can account for the observed thermal conductivity enhancement with detailed calculations. We assume that if the core-shell structure is formed, the thicknesses of the shell (TiO₂ or TiO_xS_{3-x}) and core (TiS₃) are t_1 and t_2 , respectively. Thus, the effective thermal conductance of the core-shell structure should be

$$\sigma_{eff} = \sigma_1 + \sigma_2, \quad (\text{R1})$$

where σ_1 and σ_2 are the thermal conductance of the shell and core, respectively. Since the thermal conductivity k is defined as $\sigma L/S$, where L is the length and S is the cross-sectional area, the effective thermal conductivity of the core-shell structure should be

$$k_{eff} = \frac{k_2(W-2t_1)t_2 + k_1[W(t_2+2t_1) - (W-2t_1)t_2]}{W(t_2+2t_1)}, \quad (\text{R2})$$

where W is the width of the core-shell structure, and k_1 and k_2 are the thermal conductivity of the shell and core, respectively. It should be noted that equation (R2) overestimates the effective thermal conductivity because the phonon-boundary scattering is ignored at the core-shell interface. Since the width is much larger than thickness especially for thin samples, Eq. (R2) can be simplified as

$$k_{eff} = \frac{k_2 t_2 + k_1 \cdot 2t_1}{t_2 + 2t_1}. \quad (\text{R3})$$

If we ignore the lattice contraction, for the 19 nm thick sample, the thermal conductivity k_2 of the core (TiS₃) should be smaller than that of the bulk TiS₃ (2.1 W/m-K for nanoribbons with

thickness larger than 80 nm in Fig. 2b) due to the size effect. Thus, k_2 is smaller than 2.1 W/m-K. On the other hand, the thermal conductivity of bulk TiO_2 is about 8.5 W/m-K at room temperature obtained from experimental measurement.⁵ Also due to the size effect, the value of k_1 should be smaller than 8.5 W/m-K. To solve for the maximum thermal conductivity of the core-shell structure, we took the upper limit values of k_1 as 8.5 W/m-K and k_2 as 2.1 W/m-K. Accordingly, the shell thickness t_1 dependent k_{eff} of the core-shell structure for the 19 nm thick ribbon is shown in Fig. R8b. From the figure, it is impossible to explain the measured 15.5 W/m-K thermal conductivity based on the core-shell model.

Figure R8 (a) The core-shell structure of TiS_3 . t_1 , t_2 , W and L are the thickness of shell and core, and the width and length of a core-shell structure, respectively. (b) The shell thickness dependent k_{eff} of the core-shell structure for the 19 nm thick sample.

Actually, for the large enhancement of thermal conductivity observed in the current work, the mechanism is originated from the enhanced Young's modulus induced by the lattice contraction. In order to evaluate the effects of lattice contraction on the thermal conductivity, we quantitatively calculated the Young's modulus enhancement under a 9% lattice contraction from both first-principles method (Fig. 4a) and two-body potential (see response to comment 5). Both calculations show that a 9% lattice contraction can cause a more than 6-fold increase of Young's modulus. Overall, it is the O doping induced lattice contraction rather than the presence of TiO_2 or $\text{TiO}_x\text{S}_{3-x}$ in a core-shell structure that causes the observed remarkable enhancement of thermal conductivity. These new findings could help understand thermal transport in quasi-1D vdW materials and provide a new route to modulate their thermal conductivity.

In the revised manuscript and SI, we have added a discussion about the core-shell model as follows.

Lines 297-303 in page 17 “Considering that TiO_2 has a higher thermal conductivity than TiS_3 , one might expect that formation of a core-shell (TiS_3 - $\text{TiO}_2/\text{TiO}_x\text{S}_{3-x}$) structure could explain the observed enhancement of thermal conductivity. However, detailed calculations exclude the possibility of the core-shell structure, which is described in SI. In fact, the diffraction pattern of HRTEM displays only one type of lattice structure. If a core-shell structure were formed, two types of lattice structures would be easily discerned because the weak interchain vdW interaction should allow for the TiS_3 and $\text{TiO}_2/\text{TiO}_x\text{S}_{3-x}$ layers retaining their respective lattice structures.

”

Comment #2. *TiS₃ is a quasi-1D system (can be reflected by the near linear $k \sim T$ at low temperature), meaning a -axis and c -axis are isotropic with van der Waals interaction. Then why thermal conductivity increases with width but decrease with thickness? This is related to my comment 1.*

Response: The reviewer is correct that both the couplings along the a -axis and c -axis are vdW interactions. Nevertheless, the coupling strength along the a -axis is 2.5 times stronger than that along the c -axis.⁸ Thus, the width of prepared sample is always significantly larger than the thickness along the c -axis. Actually, the minimum width for all measured samples in this work is > 50 nm. Now as the reviewer argued and we agreed, the effect is really due to O doping with a higher doping level for thinner nanoribbons, which leads to enhanced thermal conductivity for thinner ribbons. In contrast, since the width is large, the doping level (and hence lattice contraction and Young's modulus) does not vary with width. In such case, the thermal conductivity increases with width due to the reduced phonon-boundary scattering.

In the revised manuscript, we have added a discussion to explain the width and length dependent thermal conductivity as follows.

Lines 314-325 in page 18 “It is also noticed that O-doping induced lattice contraction is mainly correlated with thickness but not width and length. This is because the coupling strength along the a -axis is 2.5 times that along the c -axis.²⁶ Thus, the sample width is always significantly larger than the thickness. Actually, the minimum width for all measured samples in the current work is > 50 nm. As demonstrated in Fig.4b, the thinner ribbons have a higher doping level because it is more easy for O atoms to diffuse and replace S atoms. The O doping level starts to increase quickly as the ribbon thickness is below 50 nm. In contrast, the doping levels for all of the measured samples (and hence lattice contraction as shown in Fig. S8 (a) and Young's modulus) are not sensitive to the width variations because the minimum width for the samples is larger than 50 nm. So, the measured thermal conductivity demonstrates strong thickness dependence but relatively weak width dependence. The relatively weak width dependent thermal conductivity is attributed to the reduced phonon boundary scattering.”

Comment #3. *Figure 2a is also beyond my understanding. Thermal conductivity should increase with temperature as T^3 for 3D systems and nearly linear for 1D. However, for example, the thermal conductivity for sample with 272nm-thickness is already approaching zero, far beyond the linear $k \sim T$ curve. The thermal conductivity will be negative when temperature approaching zero.*

Response: Quasi-1D and quasi-2D systems demonstrate different temperature dependence in different temperature ranges. For example, graphite demonstrates a T^3 law below 10 K and a T^2 dependence beyond 10 K till Umklapp scattering renders more complex temperature dependence. For quasi-1D systems, there exist transition temperatures from 3D to 2D, and to 1D as temperature increases. For TiS₃, our calculations indicate that at low enough temperatures, low frequency phonons with wavevectors along all 3D directions can be excited, which implies a temperature dependence of $k \propto T^3$. With the increase of the temperature, the

activated phonons along the c -axis direction in a TiS_3 nanoribbon first approach the Brillouin zone boundary because of the low upper frequency limit as a result of the weakest coupling strength in this direction. In such case, phonon transport transitions from 3D to 2D, and the temperature dependent thermal conductivity becomes $k \propto T^2$. Based on the formula $k_B T = \hbar \omega_{max}$ and phonon dispersion (Fig. R9 (a)), the transition temperature T_{3D-2D} is estimated to be 12.2 K. With further increase in temperature, the activated phonons along the a -axis direction approach the Brillouin zone boundary, and the thermal transport becomes 1D with $k \propto T$. The transition temperature T_{2D-1D} is calculated as 25.6 K.

For the temperature around 20 K, the thermal conductivity should have a quadratic dependence on T . From Fig. R9(b), the relation of $k \propto T^2$ (blue dashed line) shows a good match with the experimental data. Thus, the thermal conductivity would not be negative when temperature approaching zero since the temperature dependence will change from 2D to 3D as the temperature further drops.

Figure R9. (a) Phonon dispersion of pristine TiS_3 . (b) Thermal conductivity of nanoribbon with different thickness. The blue dash line indicates a trend of $k \propto T^2$.

Comment #4. According to $v=(E/p)^{0.5}$, the speed of sound should also related to mass density. Since S is replace by O, and introduces lattice contraction, the mass density must change.

Response: We thank the reviewer for bringing this up. In the revised manuscript, we have recalculated the velocity considering the O substitution induced change of mass density.

Comment #5. Thermal conductivity increasement when S doped by O is quite normal since TiO_2 has much higher thermal conductivity.

Response: First of all, even with the O doping, the crystalline structure disclosed by microscopy studies is that of TiS_3 rather than TiO_2 . In addition, the measured upper limit of bulk TiO_2 thermal conductivity (8.5 W/m-K) is still lower than the value of the 19 nm thick nanoribbon sample (15.5 W/m-K). Therefore, the enhanced thermal conductivity is not a simple effect of a TiO_2 shell as we point out in the response to comment #1. Instead, our systematic study discloses a more intriguing underlying mechanism. The O doping leads to stronger interatomic interaction strength and reduced lattice constant, which correspond to a much enhanced Young's modulus. The increased Young's modulus then alters the phonon dispersion relation, leading to enhanced phonon group velocity and reduced phonon-phonon scattering rate, which render higher thermal conductivity. These physical understandings are

supported by first-principles calculations and experimental measurements as discussed in the manuscript. We would like to emphasize the effect of lattice contraction, which has not been recognized previously. Please find below a more detailed numerical analysis of the lattice contraction effect.

To demonstrate the significant effect of lattice contraction on the enhancement of thermal conductivity, we quantitatively calculated the Young's modulus change under a 9% lattice contraction from a two-body Morse potential. From a microscopic point of view, the Young's modulus can be regarded as the ratio of interatomic spring constant to the equilibrium lattice constant. Thus, the effect of lattice contraction on interatomic spring constant needs to be extracted first.

The widely accepted two-body Morse potential (V) for covalent bonds can be expressed as

$$V = D[1 - e^{-a(r-r_{min})}]^2, \quad (\text{R4})$$

where D is the potential well, a is the parameter to determine the shape of the potential, r is the distance between two interacting particles and r_{min} is the equilibrium bond length. The interatomic spring constant K , *i.e.*, the second derivative of potential energy V with respect to distance r , is

$$K = \frac{\partial^2 V}{\partial r^2} = 2Da^2[2e^{-2a(r-r_{min})} - e^{-a(r-r_{min})}]. \quad (\text{R5})$$

Thus, the value of interatomic spring constant before the lattice contraction is $K_{min} = 2Da^2$ at $r = r_{min}$. As a 9% lattice contraction occurs, the spring constant is $K_{con} = 2Da^2[2e^{0.4464a} - e^{0.2232a}]$ at $r = 0.91r_{min}$. In the calculation, the equilibrium bond distance r_{min} was taken as the Ti-S bond length along the b -axis direction. Thus, the Young's modulus is enhanced by

$$\frac{K_{con}/r_{con}}{K_{min}/r_{min}} = \frac{K_{con} r_{min}}{K_{min} r_{con}} = 2e^{0.4464a} - e^{0.2232a} \text{ times. There is no reported value of } a \text{ for Ti-S}$$

bond and we took that from Ti-O bond instead⁹. The calculated enhancement of Young's modulus is 8.7 times. The details are shown in Fig. R10.

Figure R10. The effect of lattice contraction on the Young's modulus calculated based on a two-body Morse potential.

Overall, based on the two-body Morse potential, a 9% lattice contraction leads to a significant (8.7-fold) enhancement of Young's modulus, which is consistent with the result from first-principles calculations and experimental measurements (Fig. 4a). Thus, it is the O doping induced lattice contraction rather than the presence of TiO_2 or $\text{TiO}_x\text{S}_{3-x}$ in a core-shell structure that causes the observed remarkable enhancement of thermal conductivity.

In the revised manuscript and SI, we added a discussion about the effect of lattice contraction on Young's modulus as follows.

Lines 188-191 in page 12 “Results (Fig. S10, S11) show that a 9% lattice contraction rate can induce a drastic enhancement of interatomic interaction strength and Young's modulus, which corroborate the first-principles calculations and experimental measurements (Fig. 4a).”

Reviewer #3

General remark: *Authors have presented an enhancement of the thermal conductivity of TiS_3 nanoribbons with a reduction of their thickness, and discussed its origin based on the elastic stiffening. Experimental results authors provided are of high quality and the main argument about the elastic stiffening is supported well experimentally and theoretically. However, I am not convinced that the present work delivers a message important enough to be shared in Nature Communications. As authors cited as Ref. 24, a similar experimental observation was already reported for similar quasi-1D vdW NbSe_3 nanowires. In Ref. 24, the hydraulic diameter is varied in nanometer scales, and the thermal conductivity becomes enhanced by 25 times. Authors there also determined the Young's modulus and concluded that the elastic stiffening is responsible for the thermal conductivity enhancement. In the present work, authors made similar studies including the enhancements of both the thermal conductivity and Young's modulus in the quasi-1D vdW TiS_3 nanoribbons. I found that TEM and XPS measurements were additionally conducted to further characterize lattice parameters and chemical compositions. Although authors provided two rules about dopant atoms for the thermal conductivity enhancement, I am not sure that this can guarantee a novelty for the publication in Nature Communications.*

Response: We thank the reviewer for commenting our experimental results as of high quality and well supported. We recognize that the cross-sectional size dependence of the observed thermal and mechanical properties in this manuscript are similar to that in Ref. 24. The most important contribution of this manuscript is the discovery of the underlying mechanisms that are responsible for the observed thermal and mechanical property changes, which are clearly beyond those in Ref. 24. With respect to the underlying physics, Ref. 24 stopped at the enhanced Young's modulus without any further discussion on what structural and composition changes lead to the enhancement. In contrast, in this manuscript, we characterized the O doping level, the lattice contraction induced by the O doping, and how the lattice contraction further altered the Young's modulus. We do feel that the newly disclosed underlying physics is as important as the novel thermal/mechanical trends first disclosed in Ref. 24. In addition, it is unclear whether the trends first disclosed in Ref. 24 are unique to NbSe_3 or more general for a class of quasi-1D crystals. This manuscript showed that similar size dependence can occur in

quasi-1D crystals other than NbSe₃. Lastly, the thermal conductivity trend does show some difference from that of NbSe₃. For example, while we observed a similar dependence on cross-sectional size, we did not find superdiffusive phonon transport that could lead to a length dependent thermal conductivity extending over tens of microns. Given that studies of thermal transport in quasi-1D vdW thin films/nanowires only started very recently, we do feel that the new observations and underlying mechanisms should merit a publication in *Nature Communications*.

Comment #1. *Authors often mentioned that the thickness dependence of thermal conductivity is counterintuitive when considering the size effect and confinement effect. There should be a more detailed supporting discussion for this argument.*

Response: We thank the reviewer for this suggestion. In the revised manuscript, we have added a discussion for the normal size effect and confinement effect as follows.

Lines 62-67 in page 4 “Compared with bulk TiS₃ (3.7 W/m-K), the higher thermal conductivity for the nanoribbon (4 W/m-K) is counterintuitive because the classical size effect, i.e., phonon boundary scattering, usually leads to a reduced thermal conductivity. It should be noted that the measurements of nanoribbons and bulk TiS₃ are from different groups and the slightly higher thermal conductivity of the nanoribbons may be due to sample quality, measurement error or abnormal size effect.”

Lines 116-117 in page 7 “At room temperature, κ drops from ~15.5 W/m-K for the 19 nm thick ribbon to a bulk value of ~2.1 W/m-K for ribbons of > 80 nm thick (Fig. 2(b)), which is opposite to the expectation based on the classical size effect as reported in a recent first-principles study²⁶.”

Lines 304-313 in page 17-18 “Recently, a non-monotonic thickness dependent thermal conductivity of layered TiS₃ was reported by first-principles calculations,²⁶ which suggests a minimum thermal conductivity at ~17.6 nm thick. In the current work, the thermal conductivity of TiS₃ nanoribbons starts to escalate as the thickness becomes thinner than ~80 nm and keeps the increasing trend to the thinnest measured ribbon of ~19 nm thick. We note that the non-monotonic thickness dependent thermal conductivity in Ref. 26 is for pristine TiS₃ as a result of phonon confinement purely due to size effect while the significantly enhanced thermal conductivity with decreasing thickness in this study is caused by the O substitution induced lattice contraction. This difference in the underlying mechanism leads to the different thickness dependences in Ref. 26 and current study.”

Comment #2. *In line 163, authors compared the lattice contraction rate and Young’s modulus enhancement quantitatively. There should be a further explanation why the lattice contraction rate is not enough to explain the observed enhancement of the Young’s modulus.*

Response: We are sorry for not being able to clearly show how the lattice contraction leads to a much enhanced Young’s modulus. Actually, the sentence “*the lattice contraction rate is not enough to explain the observed enhancement of the Young’s modulus*” means that we should consider not only the 9% lattice contraction but also the enhanced interatomic spring constant. Microscopically, Young’s modulus is the ratio of interatomic spring constant to the equilibrium lattice constant. In fact, the change in the interatomic spring constant due to lattice contraction

plays the dominant role in boosting the Young's modulus. To clarify this, we employed two different two-body interatomic potentials, *i.e.*, Lennard-Jones and Morse, to evaluate the effects of a 9% lattice contraction on the interatomic spring constant.

(1) Lennard-Jones potential

The widely used two-body Lennard-Jones potential V can be expressed as

$$V = 4\varepsilon\left[\left(\frac{\sigma}{r}\right)^{12} - \left(\frac{\sigma}{r}\right)^6\right], \quad (\text{R6})$$

where r is the distance between two interacting particles, ε is the depth of the potential well, and σ is the distance at which the particle-particle potential energy V is zero. The Lennard-Jones potential has its minimum at a distance of $r=r_{\min}=2^{1/6}\sigma$. Based on Eq. (R6), the interatomic spring constant K is the second derivative of potential energy V with respect to distance r :

$$K = \frac{\partial^2 V}{\partial r^2} = 4\varepsilon\left[\frac{156\sigma^{12}}{r^{14}} - \frac{42\sigma^6}{r^8}\right] \quad (\text{R7})$$

Thus, without lattice contraction, the value K_{\min} at r_{\min} is $14.287 \cdot 4\varepsilon/\sigma^2$. As a 9% lattice contraction occurs, the interatomic distance r_{con} changes to $0.91r_{\min}$, and thus, the Young's modulus K_{con} changes to $80.471 \cdot 4\varepsilon/\sigma^2$. Overall, the Young's modulus is enhanced by

$$\frac{K_{\text{con}}/r_{\text{con}}}{K_{\min}/r_{\min}} = \frac{K_{\text{con}} r_{\min}}{K_{\min} r_{\text{con}}} = 5.632 \cdot 1.099 = 6.2 \text{ times (Fig. R11), which is consistent with the}$$

results from experimental data.

Figure R11. The effect of lattice contraction on the Young's modulus calculated based on a two-body Lennard-Jones potential.

(2) Morse potential

Another widely used two-body potential is Morse, which can be expressed as:

$$V = D[1 - e^{-a(r-r_{\min})}]^2, \quad (\text{R8})$$

where D is the potential well, a is the parameter to determine the shape of the potential, r is the

distance between two interacting particles and r_{\min} is the equilibrium bond distance. The interatomic spring constant K , *i.e.*, the second derivative of potential energy V with respect to distance r , is

$$K = \frac{\partial^2 V}{\partial r^2} = 2Da^2[2e^{-2a(r-r_{\min})} - e^{-a(r-r_{\min})}], \quad (\text{R9})$$

Thus, the value of interatomic spring constant before the lattice contraction is $K_{\min} = 2Da^2$ when $r = r_{\min}$. As a 9% lattice contraction occurs, the spring constant is $K_{\text{con}} = 2Da^2[2e^{0.4464a} - e^{0.2232a}]$ at $r = 0.91r_{\min}$. In the calculation, the equilibrium bond distance r_{\min} was taken as the Ti-S bond length along the b -axis direction. Thus, the Young's modulus enhances by

$$\frac{K_{\text{con}}/r_{\text{con}}}{K_{\min}/r_{\min}} = \frac{K_{\text{con}} r_{\min}}{K_{\min} r_{\text{con}}} = 2e^{0.4464a} - e^{0.2232a} \text{ times. There is no reported value of } a \text{ for Ti-S}$$

bond and we took that from Ti-O bond instead⁹. The calculated enhancement of Young's modulus is 8.7 times. The details are shown in Fig. R12.

Figure R12. The effect of lattice contraction on the Young's modulus calculated based on a two-body Morse potential.

Overall, based on both Lennard-Jones and Morse potentials, a 9% lattice contraction leads to a significant (8.7-fold) enhancement of Young's modulus, which is consistent with the result from the first-principles calculations and experimental measurements (Fig. 4a). Thus, it is the enhanced interatomic spring constant that dominates in enhancing the Young's modulus.

In the revised manuscript and SI, we added a discussion about the effects of lattice contraction on Young's modulus from the two different two-body potentials as follows.

Lines 188-191 in page 12 "Results (Fig. S10, S11) show that a 9% lattice contraction rate can induce a drastic enhancement of interatomic interaction strength and Young's modulus, which corroborate the first-principles calculations and experimental measurements (Fig. 4a)."

Comment #3. In the discussion around line 311, how is the phonon-impurity scattering varies when F atoms are chosen as dopant atoms?

Response: We thank the reviewer for this question. In order to extract the phonon-impurity scattering cross-section and strength for TiS_3 doped with F atoms, we employed the Callaway model to fit the temperature dependent thermal conductivity of the F atom doped TiS_3 from first-principles calculations. The fitting result is shown in Fig. R13. Based on the fitting, the phonon-impurity scattering cross-section of the F atom doped TiS_3 is 47.7 times that of pristine TiS_3 and the phonon-impurity scattering strength of the F atom doped TiS_3 is five orders of magnitude larger than that of pristine TiS_3 , indicating the significant phonon-impurity scattering in the F atom doped TiS_3 . We thank the reviewer again for this comment. In the revised manuscript, we added a description of the phonon-impurity scattering in the F atom doped TiS_3 in lines 364-367 in page 20-21 with red fonts.

Figure R13. Temperature-dependent thermal conductivity of TiS_3 (red circles) and F atoms doped TiS_3 (olive rhombus). The dots and lines represent first-principles calculations and Callaway fitting results, respectively.

Comment #4. Overall, the experimental observations and the discussions have been presented quite logically. And, the conclusion is well supported. Nevertheless, the novelty of the present work does not seem to high enough, and therefore I recommend this paper would be shared in a more specialized journal.

Response: We thank the reviewer again for the positive evaluation of the quality of our work. As we discussed in the response for Comment#1, we do feel that the newly disclosed intriguing underlying physics that leads to the enhanced Young's modulus and thermal conductivity is novel enough to justify a publication in *Nature Communications*. This is particularly true give that so far, studies of thermal and mechanical properties of quasi-1D vdW crystal nanowires only started very recently and there are a lot of remaining scientific questions to answer. Particularly, contrary to the common understanding of reducing lattice thermal conductivity, doping and size confinement can actually lead to enhance lattice thermal conductivity for quasi-1D vdW crystal nanowires, which provides new routes to tune phonon transport. Importantly, the thorough study of the changes at the compositional and structural level provides in-depth

understanding for the first time, and we hope that this fact could change the reviewer's opinion on the novelty of the presented research.

Reference

1. S. Grimme, J. Antony, S. Ehrlich and H. Krieg, *J. Chem. Phys.*, **132**, 154104 (2010).
2. M. X. Sun, J. Z. Li, Q. Q. Ji, Y. X. Lin, J. T. Wang, C. Su, M. H. Chiu, Y. L. Sun, H. Y. Si, T. Palacios, J. Lu, D. Xie and J. Kong, *Phys. Rev. Mater.*, **5**, 094002 (2021).
3. A. Lipatov, M. J. Loes, H. Lu, J. Dai, P. Patoka, N. S. Vorobeva, D. S. Muratov, G. Ulrich, B. Kästner and A. Hoehl, *ACS nano*, **12**, 12713-12720 (2018).
4. K. Wu, E. Torun, H. Sahin, B. Chen, X. Fan, A. Pant, D. Parsons Wright, T. Aoki, F. M. Peeters and E. Soignard, *Nat. Commun.*, **7**, 1-7 (2016).
5. X. Feng, X. Huang and X. Wang, *Nanotechnology*, **23**, 185701 (2012).
6. D. J. Kim, D. S. Kim, S. Cho, S. W. Kim, S. H. Lee and J. C. Kim, *Int. J. Thermophys.* **25**, 281-289 (2004).
7. P. Torres and R. Rurali, *J. Phys. Chem. C*, **123**, 30851-30855 (2019).
8. C. Liu, P. Lu, D. Li, Y. Zhao and M. Hao, *Mater. Today Nano*, **17**, 100165 (2022).
9. C. Domain, C. Becquart and J. Foct, *Phys. Rev. B*, **69**, 144112 (2004).

Reviewers' comments:

Reviewer #1 (Remarks to the Author):

The authors answered questions in great detail. The authors clarified the innovation more clearly in this version. They also answered the effect of S defect in detail so that I think it can be illustrated the important role of O atoms. It seems to have a good explanation for the temperature dependent thermal conductivity. The paper is suggested to be published.

Reviewer #2 (Remarks to the Author):

I do admire author's efforts in this rebuttal since most of my comments are properly answered. However in this case my main comment will be: what's the innovation of this paper? Why the result is so special and is worthy of Nat Comm? The size effect thermal conductivity is worthy of publication since it is important in understand standing abnormal phonon transport in nanostructures. However, the result in this manuscript is not the size effect, and authors also agreed with my comment, quoting "we agree with the reviewer that the trend is not induced by size confinement". The Abstract of this manuscript is also misleading. O doping introduced lattice contraction and the corresponding increasement in Young's modulus and thermal conductivity is not new and not so import. The current results relating to doping effect is boring and this paper should be transferred to Communications Materials or Scientific Report.

Reviewer #3 (Remarks to the Author):

Authors have revised the manuscript by considering the reviews' comments especially about the novelty of their finding. However, the message delivered in the introduction does not properly address the novelty of the present work. Authors should elaborate it further by focusing more on the thermal properties of low-dimensional materials. From the beginning to the end, authors should highlight the 1D nature of the target material. Otherwise, the present work would be a simple report about the enhancement of the thermal conductivity in $\text{TiS}_{3-x}\text{O}_x$ with an increase of x.

Although authors have discussed about the phonon-impurity scattering in detail, they have not discussed the phonon-boundary scattering. Authors should show how much the phonon-boundary scattering is varied depending on the thickness. Although the boundary scattering is included in the Callaway model fitting, authors have not explained its expression in detail. Also, they have to provide the group velocity and also the boundary scattering' contribution to the thermal conductivity quantitatively depending on the thickness.

In the present work, the oxygen doping concentration was not controlled intentionally. Although one can reasonably expect that the oxygen doping can occur more easily in the thinner samples, it will be helpful if a reliable strategy to control the oxygen doping level would be commented.

Overall, the results presented in this work can provide a good strategy in enhancing the thermal conductivity in the low dimensional system. I expect that this work can be reconsidered provided that the motivation and conclusion were improved by focusing more on the thermal properties and its control in the low-dimensional materials.

Response to the Reviewers' Comments

We sincerely thank the reviewers for the valuable comments on our manuscript. We have carefully considered all comments and made revisions accordingly, as reflected in the point-by-point response below. We believe that the revisions have significantly improved the quality of the manuscript and hope the reviewers find the updated version suitable for publication.

~~~~~

### Reviewer #1

*General remark: The authors answered questions in great detail. The authors clarified the innovation more clearly in this version. They also answered the effect of S defect in detail so that I think it can be illustrated the important role of O atoms. It seems to have a good explanation for the temperature dependent thermal conductivity. The paper is suggested to be published.*

**Response:** We thank the reviewer for the agreement with the publication of current manuscript.

### Reviewer #2

*General remark: I do admire author's efforts in this rebuttal since most of my comments are properly answered. However, in this case my main comment will be: what's the innovation of this paper? Why the result is so special and is worthy of Nat Comm? The size effect thermal conductivity is worthy of publication since it is important in understanding abnormal phonon transport in nanostructures. However, the result in this manuscript is not the size effect, and authors also agreed with my comment, quoting "we agree with the reviewer that the trend is not induced by size confinement". The Abstract of this manuscript is also misleading. O doping introduced lattice contraction and the corresponding increase in Young's modulus and thermal conductivity is not new and not so import. The current results relating to doping effect is boring and this paper should be transferred to Communications Materials or Scientific Report.*

**Response:** We thank the reviewer for this concern. However, we believe that doping to enhance Young's modulus and thermal conductivity through lattice contraction is new and important. First of all, in numerous publications (e.g. *Nat. Commun.* **2019**, 10, 2814 and *NPJ Comput. Mater.* **2021**, 7, 54), it has been shown that doping in general reduces the lattice thermal conductivity through enhanced impurity scattering, which actually is a well-accepted trend in thermal science field. Importantly, doping induced lattice contraction, as a new mechanism, provides an efficient strategy to modulate phonon transport and the effects of lattice contraction on thermal conductivity has been explored only very recently by a couple of papers in very prestigious journals. For example, it is shown that in proton intercalated WO3 film, the volume contracts 3.4% and the thermal conductivity increase 1.7 times (*Adv. Mater.* **2019**, 31, 1903738). In addition, in oxygen intercalated SrCoO2.5, the lattice contracts and accompanies with a phase transition, which leads to a 2.5 times enhancement of thermal conductivity (*Nat. Mater.* **2020**, 19, 655-662). The fact that both papers are published in high impact journals very recently is a strong indication that lattice contraction induced thermal conductivity enhancement is important.

Our manuscript demonstrates that for quasi-1D van der Waals crystal nanoribbons, the lattice contraction induced up to 7 times thermal conductivity enhancement, much stronger than those reported in the above-mentioned two papers. We believe that for studies of any physical properties, a 3-fold higher enhancement than previously reported value certainly represents a

significant progress. Importantly, while the two reports mentioned the effects of lattice contraction on thermal conductivity, the underlying mechanism is not sufficiently discussed and how to employ lattice contraction to modulate phonon transport is not addressed. In contrast, our study provides a comprehensive physical picture of O-doping induced lattice contraction, and its effect on Young's modulus and lattice thermal conductivity, based on both experimental characterizations and atomistic modelling. As such, our manuscript represents a significant advance both in terms of the much higher O-doping induced thermal conductivity enhancement and the detailed atomistic understanding of the O-doping effects.

In response to the reviewer's concern, we further analyze the underlying mechanism for the enhancement of thermal conductivity in quasi-1D  $\text{TiS}_3$  since in many previous studies especially in 3D materials doping decreases the thermal conductivity despite the existence of lattice contraction such as in  $\text{Si}_{1-x}\text{Ge}_x$  (*Nano Lett.* **2008**, 8, 1, 276-280). In order to reveal the mechanism, the thermal conductivity of C doped Si (inset of Fig. R1(d)) was calculated. At a doping level of 25% ( $\text{Si}_{0.75}\text{C}_{0.25}$ ), the lattice contraction appears due to the smaller radii of C atoms. However, the thermal conductivity has a 66% decrease. Thus, it is unlikely that the increase of thermal conductivity in  $\text{TiS}_3$  is only because of doping induced lattice contraction, which should also originate from the quasi-1D vdW structure. The most distinct feature of quasi-1D vdW structure is the weak vdW interactions along the interchain directions and the presence of the vdW gaps. Then, we calculate the coupling strength along both the interchain and intrachain directions at different O doping levels (Fig. R1(a)). The coupling strength along the interchain directions fluctuates slightly, while the coupling strength along the intrachain direction demonstrates a significant increase with the O doping levels (Fig. R1(b)). It is interesting to compare the strain fields induced by doped atoms in bulk silicon and  $\text{TiS}_3$  nanoribbons. In bulk silicon, the doped C atom introduces lattice contraction along three dimensional directions due to the strong covalent bonds of C-Si. As a result, the elastic strain induced by the doped C atoms in bulk Si is along all the three dimensional directions, and the strong potential-energy and structural distortions are introduced, which significantly enhance phonon scattering and reduce phonon lifetime as demonstrated in Fig. R1(d). In contrast, when O substitutes S in a  $\text{TiS}_3$  nanoribbon, the vdW gaps provide the freedom and space for the structural evolution. As a result, the doped atoms induce the lattice contraction and enhance the coupling strength significantly only along the molecular chain direction, which leads to the significant increase of Young's modulus, phonon group velocity, and phonon lifetime (Fig. R1(c)). The increased phonon lifetime is caused by the enlarged phonon bandgap due to lattice contraction (Fig. S17).

We want to thank the reviewer again for this concern that push us to dig the novelty and underlying mechanism. In the revised manuscript, we have added the above discussion with red fonts in Fig. 5. Given the above-mentioned significant results of our work, we hope that this fact could change the reviewer's opinion on the novelty of the presented research. We understand the reviewer for the concern since the original manuscript may not describe the novelty clearly in the Abstract and Introduction. In the revised manuscript, we have made abundant revisions with red font to clarify the novelty of the current work.

Fig. R1 Comparison of mechanical and phonon transport properties between quasi-1D  $\text{TiS}_3$  doped with O atoms and bulk Si samples doped with C atoms. (a) Atomic structures of  $\text{TiS}_3$  at different O doping levels. (b) The coupling strength along the  $a$ -,  $b$ -, and  $c$ -axis directions in  $\text{TiS}_3$ . (c) Phonon frequency dependent phonon lifetime due to three-phonon scatterings in pristine and O doped  $\text{TiS}_3$ . (d) Phonon frequency dependent phonon lifetime in pristine and C doped Si.

### Reviewer #3

**General remark:** Authors have revised the manuscript by considering the reviews' comments especially about the novelty of their finding. However, the message delivered in the introduction does not properly address the novelty of the present work. Authors should elaborate it further by focusing more on the thermal properties of low-dimensional materials. From the beginning to the end, authors should highlight the 1D nature of the target material. Otherwise, the present work would be a simple report about the enhancement of the thermal conductivity in  $\text{TiS}_{3-x}\text{O}_x$  with an increase of  $x$ .

**Response:** We thank the reviewer for this suggestion. In the revised manuscript, we have added abundant revisions about the novelty of the present work with red fonts. As suggested by the reviewer, we have highlighted the 1D nature of the target material in the whole manuscript.

For example, “Quasi-1D vdW materials is composed of molecular/atomic chains with strong intrachain covalent or ionic bonds assembled through relatively weak interchain vdW interactions. Compared to the restriction of strong covalent bonds on the structure in 3D materials, the presence of the vdW gaps in quasi-1D vdW materials facilitates more freedom of structure manipulation and modification such as disassembly/reassembly12,13, ions intercalation14, and substitution15. For example, quasi-1D  $\text{Ta}_2\text{Pt}_3\text{Se}_8$  and  $\text{Ta}_2\text{Pd}_3\text{Se}_8$  nanowires can be assembled together to form the nanoscale heterojunctions13 and both layered and quasi-1D  $\text{TiS}_3$  samples have been fabricated by simply controlling the growth temperature16. Due to the existence of vdW gaps in two directions, the physical properties in quasi-1D vdW materials

should be quite different compared to those in 3D materials.”

“The replacement of sulfur atoms with small radius oxygen atoms results in significant lattice contraction and coupling strength enhancement of TiS3 along the molecular chain direction, with little effect on the vdW strengths. Compared with the doped atom inducing strain and reducing lattice thermal conductivity in 3D materials, the significant lattice contraction enhances the Young’s modulus along the molecular chain direction in thin TiS3 nanoribbons, which results in an enhanced phonon group velocity and suppressed phonon-impurity scattering strength. The combination effect leads to a 7.4-fold enhancement in thermal conductivity at room temperature. This work provides a new method to actively control phonon transport through doping low dimensional materials with small atoms.”

*Comment #1. Although authors have discussed about the phonon-impurity scattering in detail, they have not discussed the phonon-boundary scattering. Authors should show how much the phonon-boundary scattering is varied depending on the thickness. Although the boundary scattering is included in the Callaway model fitting, authors have not explained its expression in detail. Also, they have to provide the group velocity and also the boundary scattering’s contribution to the thermal conductivity quantitatively depending on the thickness.*

**Response:** We thank the reviewer for this question. Based on reference *Materials Today Nano* **2022**, 17, 100165, bulk phonon spectrum can be employed to describe the phonon properties as the thickness is larger than 20 atomic layers (17.6 nm for TiS3). In the current work, the thinnest sample (19 nm) is larger than 17.6 nm, thus,  $\tau_b^{-1} = v/1.12\sqrt{W \cdot t}$ , where  $v$  is the phonon group velocity,  $W$  is the width, and  $t$  is the thickness, is employed to consider the phonon-boundary scattering (*Physical Review B* **2002**, 66, 045302). As suggested by the formula and shown in Fig. R2, the frequency dependent phonon-boundary scattering strength increases monotonously as the thickness decreases, which is reasonable since phonons experience stronger scattering at the surface in thinner sample. In order to quantitatively evaluate the boundary scattering’s contribution to the thermal conductivity, the thermal conductivities with and without phonon-boundary scattering are compared and shown in Table R1. Clearly, with decreasing the thickness, the difference between the two cases enlarges, verifying the stronger phonon-boundary scattering in thinner sample.

Fig. R2 Phonon frequency dependent phonon-boundary scattering for different thicknesses.

Table R1. Thickness dependent thermal conductivity with and without phonon-boundary scattering at room temperature.

|                                      | 19 nm | 52 nm | 62 nm | 80 nm | 170 nm |
|--------------------------------------|-------|-------|-------|-------|--------|
| $\kappa$ without boundary scattering | 22.1  | 8.7   | 5.8   | 2.8   | 3.0    |
| $\kappa$ with boundary scattering    | 14.8  | 5.5   | 4.29  | 2.0   | 2.2    |

In order to quantitatively evaluate the group velocity's contribution, the thermal conductivities with and without lattice contraction enhanced group velocity (Young's modulus) are compared and shown in Table R2. Clearly, lattice contraction has a giant effect on the thermal conductivity especially for thinner samples. For example, for the 19 nm sample, based on the Callaway fitting, the value of thermal conductivity is 14.8 W/m-K, which drops to 2.4 W/m-K if the lattice contraction does not exist. Thus, lattice contraction enhanced phonon velocity plays a significant role in the enhancement of thermal conductivity for thinner samples.

Table R2. Thickness dependent thermal conductivity with and without lattice contraction enhanced group velocity at room temperature.

|                                  | 19 nm | 52 nm | 62 nm | 80 nm | 170 nm |
|----------------------------------|-------|-------|-------|-------|--------|
| $\kappa$ with same velocity      | 2.4   | 2.8   | 2.5   | 1.5   | 2.0    |
| $\kappa$ with different velocity | 14.8  | 5.5   | 4.29  | 2.0   | 2.2    |

We thank the reviewer for this question again. In the revised SI, we have added a discussion about the thickness dependent effects of group velocity and boundary scattering on the thermal conductivity with red fonts.

“Based on the Callaway model, the thickness dependent effects of lattice contraction enhanced group velocity (Table S5) and boundary scattering (Table S6) on the thermal conductivity are also investigated. Compared to thickness dependent boundary scattering, thickness dependent group velocity has a much stronger effect on the thermal conductivity (Table S5). For example, if the lattice contraction enhanced group velocity is ignored, the thermal conductivity of 19 nm sample is only 2.4 W/m-K and the anomalous thickness dependent thermal conductivity would disappear.”

*Comment #2. In the present work, the oxygen doping concentration was not controlled intentionally. Although one can reasonably expect that the oxygen doping can occur more easily in the thinner samples, it will be helpful if a reliable strategy to control the oxygen doping level would be commented.*

**Response:** We thank the reviewer for this suggestion. In the current work, the O atoms were introduced during the crystal growth process. As indicted by the reviewer, different oxygen doping level can be realized by choosing the appropriate thickness. However, this method is not intentional. In order to intentionally control the oxygen doping level, one possible strategy is to put the sample into an oxygen-rich environment since our first-principles calculation indicates that O atoms substituted  $\text{TiS}_3$  has a lower energy and the substitution process is energy favorable. Apart from the theoretical calculation, the possibility of putting the sample into an oxygen-rich environment to control the oxygen doping level is experimentally verified. To be more specific, from 0 to 25 h, the sample was kept in a vacuum chamber for 23 hours and then exposed to air for 2 hours. At this point, the EDS study clearly showed that the O atom concentrations on the two edges were higher than that in the center region of the  $bc$  plane. From 25 h to 49 h, the sample was placed in a vacuum chamber again and then measured the O concentration at the time point of 49 h, which indicates an enhanced and uniform O concentration across the  $bc$  plane. From 49 h to 91 h, the sample was kept in a vacuum chamber for 37 hours and placed in air for 5 hours. We observed highly accumulated O atoms near the two edges again at 91 h. The time dependent oxygen doping concentration is shown in Fig. R3. From the figure, O atoms could adsorb and then replace the S atoms in  $\text{TiS}_3$  samples although the rate for O replacing S is slow in ambient conditions. It can be inferred that the replacing rate should have an increase if the sample is put into an oxygen-rich environment. Actually, in monolayer  $\text{MoS}_2$ , it has been reported that O atoms can replace the S atoms under ambient conditions (*Nature Chemistry* **2018**, 10, 1246-1251). Overall, the oxygen doping level in  $\text{TiS}_3$  can be intentionally controlled. We thank the reviewer for this suggestion again and in the revised manuscript we have discussed the method to intentionally control the oxygen doping level.

Fig. R3 The evolution of O atoms concentration in the *bc* plane with the elapse of time.

“Finally, we want to briefly discuss how to intentionally control the oxygen doping level in  $\text{TiS}_3$  since the doping level in the current study is controlled by nanoribbon thickness, which is not intentional. As discussed above, the atomic modelling indicates that the O atom doped  $\text{TiS}_3$  has a lower energy compared to pristine  $\text{TiS}_3$ , indicating that the substitution of S by O atoms is energy favorable. In order to verify the modelling result, the evolution of O atom concentration with the elapse of time is experimentally measured and the details are shown in Fig. S19 in the SI, which demonstrates that O atoms could adsorb and then replace S atoms in  $\text{TiS}_3$  samples though the substitution rate is rather slow in ambient conditions as shown in Fig. 6(b). However, the substitution of S by O atoms is spontaneous in  $\text{TiS}_3$ , which is similar to that the O atoms can replace S atoms in monolayer  $\text{MoS}_2$ 47. If the sample is put into an oxygen-rich environment, the substitution rate should be high, which may provide an intentional method to control the oxygen doping level.”

**Comment #3.** Overall, the results presented in this work can provide a good strategy in enhancing the thermal conductivity in the low dimensional system. I expect that this work can be reconsidered provided that the motivation and conclusion were improved by focusing more on the thermal properties and its control in the low-dimensional materials.

**Response:** We thank the reviewer for the positive evaluation of our work. In the revised manuscript, we have added abundant revisions to improve the motivation and conclusion by focusing more on the thermal properties and its control in the low-dimensional materials with red fonts.

For example, “As we know, doping is a widely adopted and effective strategy to reduce thermal conductivity in 3D materials, and the mechanism of the reduced thermal conductivity is attributed to the enhanced phonon-impurity scattering strength. It is curious that doping leads to an increase in the thermal conductivity for the quasi-1D vdW materials. In order to reveal the underlying mechanism, the thermal conductivity of a bulk Si sample doped with C atoms (inset of Fig. 5(d)) is calculated. At a doping level of 25% ( $\text{Si}_{0.75}\text{C}_{0.25}$ ), the lattice contraction appears due to the smaller radii of C atoms. However, the thermal conductivity decreases to

Fig. 5 Comparison of mechanical and phonon transport properties between quasi-1D  $\text{TiS}_3$  doped with O atoms and bulk Si samples doped with C atoms. (a) Atomic structures of  $\text{TiS}_3$  at different O doping levels. (b) The coupling strength along the  $a$ -,  $b$ -, and  $c$ -axis directions in  $\text{TiS}_3$ . (c) Phonon frequency dependent phonon lifetime due to three-phonon scatterings in pristine and O doped  $\text{TiS}_3$ . (d) Phonon frequency dependent phonon lifetime in pristine and C doped Si.

34% of the pure Si sample. Thus, it is unlikely that the increase of thermal conductivity in  $\text{TiS}_3$  is only because of doping induced lattice contraction, which should also originate from the quasi-1D vdW structure. The most distinct feature of quasi-1D vdW structure is the weak vdW interactions along the interchain directions and the presence of the vdW gaps. Based on the above knowledge, we calculate the coupling strength along both the interchain and intrachain directions at different O doping levels (Fig. 5(a)). The coupling strength along the interchain directions fluctuates slightly, while the coupling strength along the intrachain direction demonstrates a significant increase with the O doping levels (Fig. 5(b)). It is interesting to compare the strain fields induced by doped atoms in bulk silicon and  $\text{TiS}_3$  nanoribbons. In bulk silicon, the doped C atom introduces lattice contraction along three dimensional directions due to the strong covalent bonds of C-Si. As a result, the elastic strain induced by the doped C atoms in bulk Si is along all the three dimensional directions, and the strong potential-energy and structural distortions are introduced, which significantly enhance phonon scattering and reduce phonon lifetime as demonstrated in Fig. 5(d). In contrast, when O substitutes S in a  $\text{TiS}_3$  nanoribbon, the vdW gaps provide the freedom and space for the structural evolution. As a result, the doped atoms induce the lattice contraction and enhance the coupling strength significantly only along the molecular chain direction, which leads to the significant increase of Young's modulus, phonon group velocity, and phonon lifetime (Fig. 5(c)). The increased phonon lifetime is caused by the enlarged phonon bandgap due to lattice contraction (Fig. S17)."

## REVIEWERS' COMMENTS:

### Reviewer #2 (Remarks to the Author):

In this new version author calculated the thermal conductivity of  $\text{TiS}_3$ ,  $\text{TiS}_2\text{O}$ ,  $\text{TiSO}_2$  and  $\text{TiO}_3$  and found thermal conductivity increase gradually due to the increased phonon lifetime. The physical picture is quite clear that with increased O, thermal conductivity of  $\text{TiS}_{3-x}\text{O}_x$  increase since thermal conductivity of  $\text{TiO}_3$  is much higher  $\text{TiS}_3$ . This is quite straightforward and NORMAL physics and hundreds of examples can be found, therefore i am quite confused with the title "abnormal".

The situation gets awkward here since in my previous report I commented that "I do believe that thermal conductivity increase with O doping since  $\text{TiO}_2$  have much higher thermal conductivity. " and authors disagree and said that thermal conductivity of  $\text{TiO}_2$  is much lower than the measured value in this manuscript, quoting "First of all, the thermal conductivity of  $\text{TiO}_2$  at room temperature has been experimentally measured and theoretically predicted with the values ranging from 1.3 to 15 W/m-K, among which the values for nanoribbons are lower than the bulk value due to phonon-boundary scattering (Table R2). Importantly, even the highest value of  $\text{TiO}_2$  is smaller than the  $\sim 15.5$  W/m-K for the 19 nm thick nanoribbon sample in our measurement." I might confused  $\text{TiO}_2$  with  $\text{TiO}_3$  but the figure 5 in the current version seems to support my previous comment.

### Reviewer #3 (Remarks to the Author):

In the revised manuscript, I found that the novelty of the present work is presented with an appropriate emphasis on the thermal conductivity enhancement in 1D vdW crystals. Although the size confinement effect was not the main issue, the size reduction could lead to the anomalous doping effects which are attributed to the one-dimensional vdW nature of the crystal. Since the authors have replied to reviewers' comments reasonably well and also improved the description about the novelty of the present work, I now recommend the publication of this work in Nature Communications.

## Response to the Reviewers' Comments

We sincerely thank the reviewers for the valuable comments on our manuscript. We have carefully considered all comments and made revisions accordingly, as reflected in the point-by-point response below.

~~~~~

Reviewer #2

Comment #1: *In this new version author calculated the thermal conductivity of TiS_3 , Ti_2O , TiSO_2 and TiO_3 and found thermal conductivity increase gradually due to the increased phonon lifetime. The physical picture is quite clear that with increased O, thermal conductivity of $\text{TiS}_{3-x}\text{O}_x$ increase since thermal conductivity of TiO_3 is much higher TiS_3 . This is quite straightforward and NORMAL physics and hundreds of examples can be found, therefore i am quite confused with the title "abnormal".*

Response: We thank the reviewer for this suggestion. In the revised manuscript, we have deleted the word “abnormal” or changed the word “abnormal” to “unexpected”.

Comment #2: *The situation gets awkward here since in my previous report I commented that "I do believe that thermal conductivity increase with O doping since TiO_2 have much higher thermal conductivity. " and authors disagree and said that thermal conductivity of TiO_2 is much lower than the measured value in this manuscript, quoting "First of all, the thermal conductivity of TiO_2 at room temperature has been experimentally measured and theoretically predicted with the values ranging from 1.3 to 15 W/m-K, among which the values for nanoribbons are lower than the bulk value due to phonon-boundary scattering (Table R2). Importantly, even the highest value of TiO_2 is smaller than the ~15.5 W/m-K for the 19 nm thick nanoribbon sample in our measurement." I might confused TiO_2 with TiO_3 but the figure 5 in the current version seems to support my previous comment.*

Response: We thank the reviewer for this insightful comment. Though TiO_3 has a higher thermal conductivity than TiS_3 , the thermal conductivity increase in O doped TiS_3 is still beyond our common sense. The doping cases for C-doped Si and O-doped TiS_3 are strikingly similar because the dopant atoms, when compared to the host atoms, belong to the same group in the periodic table and possess a smaller atomic radius. If the enhancement in thermal conductivity observed in O-doped TiS_3 is caused by the higher intrinsic thermal conductivity of TiO_2 or TiO_3 than TiS_3 , the thermal conductivity of silicon doped with C should be higher than that of pure silicon, because the thermal conductivity of diamond (C) is much greater than silicon. However, as shown in Fig. 5, the thermal conductivity of Si doped with C has a 66% drop compared to pure Si. Actually, the widely accepted knowledge is that doping decreases rather than increases the thermal conductivity because the dopant atoms distort the lattice structure and enhance the phonon scatterings. In our manuscript, the enhancement in thermal conductivity observed in O-doped TiS_3 primarily arises from its quasi-one-dimensional structure and the lattice contraction induced by doping, rather than the higher intrinsic thermal conductivity of TiO_2 or TiO_3 than TiS_3 . Our research provides a novel route for manipulating phonon thermal conductivity using low-dimensional materials.

~~~~~

**Reviewer #3**

*General remark: In the revised manuscript, I found that the novelty of the present work is presented with an appropriate emphasis on the thermal conductivity enhancement in 1D vdW crystals. Although the size confinement effect was not the main issue, the size reduction could lead to the anomalous doping effects which are attributed to the one-dimensional vdW nature of the crystal. Since the authors have replied to reviewers' comments reasonably well and also improved the description about the novelty of the present work, I now recommend the publication of this work in Nature Communications.*

**Response:** We are grateful for the reviewer's time and effort in reviewing our manuscript. We sincerely appreciate the reviewer's endorsement for the publication of our manuscript.